# Swap-and-Spoil: Untargeted Byzantine Attacks via Class-Consistent View Swaps in Vertical Federated Learning

## Abstract

Vertical Federated Learning (VFL) secures a highly privacy-preserving multi-party training paradigm in which features are vertically distributed across participants for the same sample space. Security attacks against VFL have gained attention recently, but most discussions revolve around data poisoning attacks such as backdoor attacks. Byzantine attack against a federated learning system can target the main model performance and drop its accuracy with a single adversary participating in the training. While such untargeted Byzantine attacks have been explored in horizontal settings, they still remain underexplored in vertical settings of federated systems. In this paper, we demonstrate how an adversary can mount a successful untargeted Byzantine attack that drives down the global model's inference-time accuracy. To realize this, we perform a consistent cluster-based swapping in the feature space, creating a persistent and poisoned cross-view association during training. The model internalizes this adversary-induced association and, when evaluated on clean, correctly aligned data, fails dramatically. We also show that, the widely-practiced defenses in VFL fail to detect the attack without degrading the model performance. Through this endeavour, our findings establish untargeted Byzantine attacks as a real, underexplored threat to VFL and motivate the design of robust, VFL-specific defenses.

## 1 Introduction

Federated learning (FL) emerged from the necessity of multiple organizations to collaboratively train a model without centralizing raw data, thereby reducing regulatory and operational barriers associated with data sharing (McMahan et al., 2023). In its most common form of *horizontal* FL (HFL), participants hold disjoint samples but share the same feature space; model updates are aggregated across clients, typically through variants of FedAvg (McMahan et al., 2023). In contrast, *vertical* FL (VFL) partitions the *feature* space across parties that cover an overlapping set of entities, for example a bank and a telecom serving the same customers.(Vepakomma et al., 2018). VFL serves high–value use cases where cross–silo collaboration is necessary to combine complementary attributes, such as credit scoring, risk modeling, and healthcare analytics (Yang et al., 2023a). However, the very design that makes VFL attractive, which is the feature complementarity with only one party holding labels, also introduces unique attack surfaces and a different trust boundary than in HFL.

Whereas HFL protocol involves sending local model weights with the central server in each update, VFL training operates by exchanging intermediate embeddings from each party's bottom model and training a global model at a server that typically owns the label space (Cheng et al., 2021). Even though this configuration serves the purpose of securing private data and enabling collaborative training just by owning a subset of features, keeping labels at a single site and participation of multiple parties create a scope of systematic corruption where one party's embeddings can reshape the joint decision surface for *all* entities, even when other parties behave honestly. Moreover, gradient signals sent back from the server to bottom models may leak sensitive information or be manipulated to steer local training (Fu et al., 2022; Jin et al., 2021). As a result, defending VFL demands tools beyond the HFL toolbox for robust aggregation and anomaly detection.

There's been several studies that focus on threats to VFL at training phase as well as the inference phase (e.g., (Naseri et al., 2023; Fu et al., 2022; Yang et al., 2023b; Liu et al., 2024; Zou et al., 2022)). Inference phase attacks typically leave the trained model intact and once the attack vector is removed, the deployed system can recover without retraining. In contrast, training phase attacks (He et al., 2024; Chen et al., 2024; Gu & Bai, 2023) compromises contaminate the learned parameters themselves. Once the server has been fit on corrupted embeddings, the resulting global model is intrinsically flawed, forcing an expensive retraining cycle and prolonged loss of service. For that reason, we focus on a training-phase threat that will compromise the performance of the global model once trained.

VFL's configuration of clients sending smashed representations and the server returnig gradients—expose two high stake surfaces. First, *label inference* shows that gradients or intermediate activations can reveal the server's labels to a label-blind party (Fu et al., 2022; He et al., 2024). Second, *poisoning* attacks use crafted inputs or embedding perturbations to skew the global decision boundary during training phase. In parallel, the literature on HFL attacks distinguishes a stronger class of *Byzantine* behavior, where a client acts arbitrarily and persistently malicious, not merely injecting a sparse backdoor but pushing the joint model off course (Fang et al., 2020). Unlike classic backdoors, which try to misclassify specific trigger–class pairs while preserving clean accuracy, Byzantine objectives are *untargeted*: they aim to degrade overall accuracy while remaining hard to detect throughout training.

**Why HFL defenses do not directly transfer.** Since Byzantine attack is a widely studied attack in HFL, it remains underexplored in VFL with not so many mitigations. Many defenses in HFL rely on the fact that benign clients train on the same feature space and therefore produce updates with similar statistics; robust aggregators can then down-weight outliers ((Shi et al., 2022; So et al., 2020)) or certify convergence under bounded Byzantine fractions. In VFL, however, the server does not aggregate homogeneous parameter deltas. Rather, it combines *representations from different feature spaces*. Consequently, "outlier among models" is ill-defined, and filtering based on update similarity risks discarding legitimate parties that are simply different by construction. Several VFL-specific detectors therefore monitor smashed-feature distributions, temporal gradient similarity, or reconstruction errors to flag anomalies. Yet these mechanisms are largely calibrated for *i.i.d.* or batch-local deviations and struggle against attacks that introduce *globally consistent*, slowly drifting corruption.

**State of Byzantine attacks in VFL and open gaps.** Security assessments of Byzantine VFL have been explored with and without inter-client communication, e.g., Xu et al. (2024) and Yuan et al. (2022), respectively. The latter line analyzes $l_2$-regularized finite-sum objectives and proposes dual-space defenses under a no-communication setting. While valuable, two practical gaps remain. First, attacks that tamper with values en route to the server may be neutralized by attaching a message authentication code (MAC) to client-to-server payloads; if the MAC is computed inside a trusted execution environment (TEE), post-hoc manipulation becomes implausible in many enterprise deployments. Hence, credible evaluations should assume tamper-evident channels and focus on attacks that look *statistically legitimate* to the server. Second, existing experiments often use simplified synthetic data and shallow models; it is unclear whether the same defenses scale to real multi-modal workloads where class structure, representation geometry, and training dynamics are substantially more complex. Moreover, dual reformulations that help in linear settings can become unwieldy—or intractable—once deep models are involved, motivating lighter-weight, protocol-compatible defenses that do not rely on dual optimization.

**Our Contribution** We study a training-phase, untargeted Byzantine threat tailored to the realities above: the attacker never tampers with the transport channel, needs only a *small* auxiliary labeled slice to recover latent class structure from its own feature view, and then injects a *globally consistent* corruption that remains batch-wise plausible yet collapses clean accuracy after training. Contrasting with prior VFL backdoors (Naseri et al., 2023; He et al., 2024) and label-inference attacks (Fu et al., 2022), our objective is not to stealthily implant triggers or steal labels, but to *re-wire* cross-view associations learned by the head model. We also show why common VFL detectors like grad-normal clipping, per-label smashed-feature drift checks, and AE-based anomaly scores, can admit such consistent shifts, underscoring the need for defenses that reason about *global* representation alignment rather than local deviation alone.

## 2 PRELIMINARIES

### 2.1 VERTICAL FEDERATED LEARNING

VFL enables multiple parties to collaboratively train a machine learning model by vertically sharing the feature set of same sample space. Each party holds a different subset of features privately owned by them for a common set of entities. Only one active client or a third party server typically has access to the label space and trains the global model by aggregating the embeddings received from passive clients. Let there be $K$ clients, indexed by $c \in \{1, 2, \ldots, K\}$, and a set of $N$ aligned samples. For sample $i \in \{1, \ldots, N\}$, the full feature vector is denoted as $x_i = \{x_i^{(1)}, x_i^{(2)}, \ldots, x_i^{(K)}\}$, where $x_i^{(c)} \in \mathbb{R}^{d_c}$ is the local view owned by client $c$, and the total dimensionality is $d = \sum_c d_c$. Each client $c$ holds a local *bottom model* $f_c(\cdot; \theta_c)$, which encodes its local view into an embedding: $h_i^{(c)} = f_c(x_i^{(c)}; \theta_c) \in \mathbb{R}^{p_c}$. The embeddings $\{h_i^{(c)}\}_{c=1}^{K}$ are sent to the server, which concatenates them and applies a *head model* $g(\cdot; \theta_g)$ to produce a prediction:

$$\hat{y}_i = g\left(\|_{c=1}^{K} h_i^{(c)}; \theta_g\right),$$

Only the active client or server holds labels $y_i \in \mathcal{Y}$. The end-to-end model is optimized by minimizing:

$$\min_{\Theta} \frac{1}{N} \sum_{i=1}^{N} \mathcal{L}(\hat{y}_i, y_i), \quad \text{where } \Theta = \{\theta_1, \ldots, \theta_K, \theta_g\}.$$

The backward pass proceeds in reverse: the server computes $\nabla_{h_i^{(c)}} \mathcal{L}$ for each client and sends it back, enabling each client to update its local model without learning others' raw inputs or gradients directly. This protocol enables the collaborative training of a unified model on the joint feature space while confining data exposure to intermediate latent representations.

### 2.2 BYZANTINE ATTACK IN VFL

Byzantine attacks in VFL aim towards adversarial manipulations carried out by a subset of malicious clients $\mathcal{B} \subset \{1, \ldots, K\}$ on their subset of features, who intentionally deviate from the training protocol to disrupt the global model's performance. These adversaries aim to degrade accuracy or inject targeted behavior into the model while remaining undetected by the aggregation mechanism. In HFL, each Byzantine client $c \in \mathcal{B}$ locally replaces its clean bottom model $f_c(\cdot; \theta_c)$ with an adversarial variant $f_c^{\mathcal{B}}(\cdot; \theta_c^{\mathcal{B}})$, generating corrupted embeddings: $\tilde{h}_i^{(c)} = f_c^{\mathcal{B}}(x_i^{(c)}; \theta_c^{\mathcal{B}})$, for $c \in \mathcal{B}$. In VFL, the adversarial space differs fundamentally from Horizontal FL, as clients do not exchange model weights but instead transmit intermediate embeddings. Since the data distribution is feature-partitioned, these clients can either manipulate their local input features $x_i^{(c)}$, or tamper with the generated embeddings $h_i^{(c)}$, without deviating from the standard communication protocol. Each client $c \in \mathcal{B}$ retains its original bottom model $f_c(\cdot; \theta_c)$, but generates corrupted embeddings by either poisoning the local input features or perturbing the output embeddings:

$$\tilde{h}_i^{(c)} = f_c(\tilde{x}_i^{(c)}; \theta_c) \quad \text{or} \quad \tilde{h}_i^{(c)} = \psi\left(f_c(x_i^{(c)}; \theta_c)\right),$$

where $\tilde{x}_i^{(c)}$ is a maliciously crafted version of the clean input $x_i^{(c)}$, and $\psi(\cdot)$ denotes an embedding-level transformation applied post-encoding.

Let the aggregated representation be:

$$\tilde{H}_i = \begin{cases} \tilde{h}_i^{(c)} & \text{if } c \in \mathcal{B}, \\ h_i^{(c)} & \text{otherwise,} \end{cases} \quad \text{so that} \quad \hat{y}_i = g\left(\|_{c=1}^{K} \tilde{h}_i^{(c)}; \theta_g\right).$$

While benign clients continue to minimize the standard supervised objective,

$$\min_{\Theta \setminus \{\theta_c^{\mathcal{B}}\}} \frac{1}{N} \sum_{i=1}^{N} \mathcal{L}(\hat{y}_i, y_i),$$

the Byzantine clients optimize a separate adversarial loss:

$$\min_{\theta_c^{\mathcal{B}}} \frac{1}{N} \sum_{i=1}^{N} \mathcal{L}_{\text{adv}} \left( g \left( \|_{c=1}^{K} \tilde{h}_i^{(c)} \right), y_i' \right),$$

where $y_i' \in \mathcal{Y}$ is a malicious target label in the case of a targeted attack, or an incorrect class in the case of untargeted disruption. The goal is to induce harmful behavior on poisoned inputs without compromising the model's clean performance, thereby achieving stealth.

### THREAT MODEL

In this work, we adopt a widely-used and practically relevant two-party VFL setting which is especially more vulnerable to Byzantine attacks as one client controls half of the feature space and can influence the aggregated representations sent to the server. We consider a threat model where an adversary fully compromises one of the passive clients, controlling both its private dataset and local bottom model. The passive adversary doesn't have any control over the feature subset or model parameters held by other benign clients, nor the label space maintained by the server. The attacker doesn't have any control over the global model or server-side optimization either.

To simulate a realistic adversarial capacity, we assume that the malicious client has access to a small auxiliary dataset, consisting of 5% of the training samples for which it possesses both its own feature subset and the corresponding labels. This reflects practical scenarios where cross-organization leakage, public records, or historical supervision may reveal partial labels to otherwise label-blind parties. Such assumptions are also made in prior studies (Fu et al., 2022), where limited supervision is leveraged to infer or approximate the global task.

**Adversarial Objective.** In this work, the passive malicious client seeks to degrade the overall utility of the global VFL model by injecting embeddings that create a persistent, semantically incorrect cross-view association during training rather than aiming for any targeted misclassification or trigger-based behaviour. The objective is therefore *untargeted*: the adversary attempts to corrupt the learned decision boundary so that the final model performs poorly on clean, correctly aligned data. Formally, let $P_{\text{train}}$ denote the clean joint distribution over aligned representations $(h^{(1)}, h^{(2)}, y)$. The adversary replaces the left-view inputs with a cluster-swapped variant $\widetilde{X}^{(1)}$, inducing a corrupted training distribution

$$P_{\text{adv}} = \left( \widetilde{h}^{(1)}, h^{(2)}, y \right), \qquad \widetilde{h}^{(1)} = f_1(\widetilde{X}^{(1)}).$$

The server then learns

$$g^\star = \arg\min_g \mathbb{E}_{(\widetilde{h}^{(1)}, h^{(2)}, y) \sim P_{\text{adv}}} \mathcal{L}\left( g(\widetilde{h}^{(1)}, h^{(2)}), y \right),$$

The adversary therefore seeks a corrupted view $\widetilde{X}^{(1)}$ that maximizes the mismatch between the training and test distributions:

$$\max_{\widetilde{X}^{(1)}} \text{Dist}(P_{\text{adv}}, P_{\text{train}}),$$

forcing $g^\star$ to internalize an incorrect cross-view association and leading to substantial performance degradation at inference.

## 3 METHODOLOGY

To achieve the Byzantine objective, we design a attack pipeline with two stages. The first stage constructs high-fidelity clusters of the training samples without observing the true labels. In the second stage, these inferred clusters guide a consistent feature-swapping attack in the training phase that distorts the association between feature subsets during vertical training.

### 3.1 STAGE I: CLUSTERING VIA LATENT REPRESENTATION

Since the adversary is a passive client in a vertically partitioned federation, it observes only its own feature subset $x_i^{(a)}$ for each aligned sample $i$ and never sees the true label $y_i$. To partition the

sample space into groups that approximate the hidden label distribution, the adversary leverages a small auxiliary dataset $\mathcal{D}_L$ containing approximately $5\%$ of the training entities for which both its own features and the corresponding labels are available. Such partial supervision is realistic in cross-organization collaborations where records may overlap across public registries or historical agreements, and similar assumptions have been adopted in prior vertical-FL security studies.

Let $\{x_i^{(a)}\}_{i=1}^N$ denote the adversary's local views of the $N$ aligned entities. The goal is to learn a representation function

$$\phi: \ x_i^{(a)} \mapsto z_i \in \mathbb{R}^p$$

such that the embeddings $z_i$ exhibit high intra-class similarity and inter-class separation even when the class labels are unknown for most samples.

**Phase I: Unsupervised SimCLR.** To obtain such representations we adopt a *two-phase* contrastive learning framework. In the first phase, an encoder $e(\cdot; \theta)$ is trained with the SimCLR objective on the unlabeled portion of the dataset. For each sample $x_i^{(a)}$, two stochastic augmentations $t_1, t_2 \sim \mathcal{T}$ (including random rotations, translations, brightness jitter, and occasional Gaussian blur) generate positive pairs $\big(t_1(x_i^{(a)}), \ t_2(x_i^{(a)})\big)$. The encoder parameters are optimized to minimize the normalized temperature–scaled cross-entropy (NT-Xent) loss

$$\mathcal{L}_{\text{SimCLR}} = -\frac{1}{B} \sum_{i=1}^B \log \frac{\exp\Big(\text{sim}\big(e(t_1(x_i^{(a)})), \ e(t_2(x_i^{(a)}))\big)/\tau\Big)}{\sum_{k \neq i} \exp\Big(\text{sim}\big(e(t_1(x_i^{(a)})), \ e(t_2(x_k^{(a)}))\big)/\tau\Big)},$$

where $\text{sim}(\cdot, \cdot)$ denotes cosine similarity, $B$ is the batch size, and $\tau$ is a temperature parameter controlling concentration.

**Phase II: Supervised Contrastive Refinement.** The unsupervised encoder produces latent features that capture high-level semantic structure but may still suffer from class-cluster ambiguity. Therefore, in the second phase we refine the encoder using the small labeled subset $\mathcal{D}_L$ through *supervised contrastive learning*. Given two augmented views $u, v$ of a labeled sample and the label set $\mathcal{Y}$, the supervised contrastive loss

$$\mathcal{L}_{\text{SupCon}} = -\sum_{i \in \mathcal{D}_L} \frac{1}{|\mathcal{P}(i)|} \sum_{p \in \mathcal{P}(i)} \log \frac{\exp\Big(\text{sim}\big(e(u_i), \ e(u_p)\big)/\tau\Big)}{\sum_{a \neq i} \exp\Big(\text{sim}\big(e(u_i), \ e(u_a)\big)/\tau\Big)},$$

encourages embeddings of samples with the same label to remain close while pushing apart those with different labels; here $\mathcal{P}(i)$ is the set of positives sharing the same label as sample $i$.

**Clustering by GMM and Confidence.** After contrastive pre-training and fine-tuning, each sample is embedded as $z_i = e(x_i^{(a)}; \theta^*)$. We L2-normalize these embeddings and apply a Gaussian Mixture Model (GMM) with diagonal covariance to partition the space into $K$ clusters:

$$\{z_i\}_{i=1}^N \xrightarrow{\ \text{GMM}\ } \{c_i\}_{i=1}^N, \qquad c_i \in \{1, \dots, K\}.$$

The GMM provides soft membership probabilities (responsibilities), allowing the adversary to retain cluster-confidence scores for later use in the attack.

This hybrid *SimCLR–SupCon–GMM* pipeline consistently outperforms classical alternatives such as raw-pixel KMeans, HOG-feature clustering, or denoising autoencoders. For real-world datasets, where the visual signal carried by a single vertical view is often too weak to produce well-separated contrastive embeddings, we implement an additional mechanism. To handle this low-information regime, we additionally employ a *FixMatch* framework with an exponential-moving-average (EMA) teacher to generate cluster assignments. A student network is trained with a small labeled subset and a large pool of unlabeled samples using weak–strong augmentation consistency, while the EMA teacher supplies pseudo-labels for high-confidence predictions. The teacher parameters are updated by momentum averaging,

$$\theta_t \leftarrow m\, \theta_t + (1-m)\, \theta_s,$$

where $\theta_s$ and $\theta_t$ are the student and teacher parameters and $m \in [0, 1)$ is the EMA coefficient. After training, the cluster identity of sample $i$ is taken as

$$c_i = \arg\max \mathrm{softmax}\big(f_t(x_i^{(a)})\big),$$

and the associated softmax maximum provides the cluster confidence. This teacher-driven partitioning yields sharper decision boundaries than contrastive clustering when the adversary's half-view lacks sufficient semantics, while producing the same cluster assignments required for the subsequent feature-swapping stage.

Empirically it achieves higher purity, normalized mutual information, and adjusted Rand index, yielding clusters that closely mirror the unknown label partition and thus supply a high-quality surrogate for guiding the subsequent feature–swapping attack.

### 3.2 Stage II: Consistent Cluster–Swap Poisoning

**Intuition and training-time mechanism.** Guided by the predicted clusters from Stage I, the adversary constructs a poisoned training set by *consistently* exchanging left-view features across clusters. The core intuition is that in vertical federated learning, the global model relies on stable cross-party correlations between the feature subsets of each entity. If these correlations are systematically disrupted during training, the global decision boundary will be driven toward spurious associations that do not generalize to clean data. Let the cluster partition be $\mathcal{G} = \{C_1, \ldots, C_K\}$ over the index set $[N]$, and let the adversary-controlled view be $X_A = \{x_i^{(A)}\}_{i=1}^N$, with the honest view $X_B = \{x_i^{(B)}\}_{i=1}^N$ and labels $Y = \{y_i\}_{i=1}^N$ held by the server. For each cluster $C_s$, the attacker selects a *destination* cluster $C_{t(s)} \neq C_s$ and replaces *all* left-view samples in $C_s$ with donors drawn from $C_{t(s)}$. Writing $c(i) \in \{1, \ldots, K\}$ for the cluster identity of sample $i$, the swap is

$$\tilde{x}_i^{(A)} \leftarrow x_{j(i)}^{(A)} \quad \text{with} \quad j(i) \in C_{t(c(i))}, \qquad \forall\, i \in [N],$$

and is implemented so that every $i$ receives a nontrivial donor $j(i) \neq i$ and no source cluster maps to itself. This produces a *cluster-consistent* but *semantically inverted* coupling between the adversary's left-view and the server's clean right-view: during training, the model repeatedly observes $(\tilde{x}_i^{(A)}, x_i^{(B)})$ whose two halves are drawn from different label-structured manifolds. The server then minimizes the empirical risk on poisoned tuples,

$$\min_{\Theta} \; \frac{1}{N} \sum_{i=1}^N \mathcal{L}\Big(g\big(\|_{c=1}^K h_i^{(c)}\big), y_i\Big) \quad \text{with} \quad h_i^{(A)} = f_A\big(\tilde{x}_i^{(A)}; \theta_A\big),$$

thereby learning a *stable but false* cross-view association (left-view features from the destination cluster paired with right-view features from the source cluster). At inference time, when clean tuples $\big(x_i^{(A)}, x_i^{(B)}\big)$ are presented, this association is violated, and the model's decision rule misaligns with the true joint distribution, leading to a marked accuracy drop.

**Consistent mappings as cluster-level permutations.** The consistency of the attack is captured by a derangement $\pi$ over clusters, $\pi \in \mathfrak{D}_K = \{\text{permutations of } [K] \text{ with no fixed point}\}$, so that $t(s) = \pi(s)$ and

$$\forall\, s \in [K], \; \forall\, i \in C_s: \qquad \tilde{x}_i^{(A)} \in \{x_j^{(A)} : j \in C_{\pi(s)}\}, \quad \pi(s) \neq s.$$

This ensures that every training exposure from $C_s$ consistently carries the left-view statistics of $C_{\pi(s)}$, maximizing the model's incentive to internalize the spurious cross-view coupling.

**Algorithmic feature-swap strategies.** Having fixed the partition $\mathcal{G}$, we instantiate $\mathcal{S}(X_A; \mathcal{G})$ via several pairing rules, ordered by strength.

*Optimal top-$k$ farthest swap.* We compute L2-normalized cluster centroids in the adversary's view, $\mu_s = \frac{1}{|C_s|} \sum_{i \in C_s} v_i$, $v_i = \frac{x_i^{(A)}}{\|x_i^{(A)}\|_2}$, and the inter-centroid cosine distance matrix $D_{st} = 1 - \langle \mu_s, \mu_t \rangle$. We then solve a maximum-distance derangement:

$$\pi^\star \in \arg\max_{\pi \in \mathfrak{D}_K} \sum_{s=1}^K D_{s, \pi(s)}.$$

---

**Algorithm 1** Class-Consistent View Swapping (CCVS)

---

**Require:** Adversarial views $\{x_i^{(a)}\}_{i=1}^N$, cluster ids $g_i \in \{1, \ldots, K\}$, embeddings $v_i = \phi(x_i^{(a)})/\|\phi(x_i^{(a)})\|_2$, top-$k$ target clusters $T(g)$.

**Ensure:** Swapped views $\{\tilde{x}_i^{(a)}\}$.

1: For each $g$, set $I_g = \{i : g_i = g\}$ and compute centroid $\mu_g = \frac{1}{|I_g|} \sum_{i \in I_g} v_i$ (L2-normalized).
2: For each $g$, define $T(g)$ as the $k$ clusters with largest cosine distance $1 - \langle \mu_g, \mu_h \rangle$.
3: **for** $s = 1$ to $K$ **do**
4:    **for** $i \in I_s$ **do**
5:       Sample target cluster $t \sim T(s)$ uniformly.
6:       Choose donor
$$j^\star = \arg\min_{j \in I_t} \langle v_i, v_j \rangle$$
7:       Set $\tilde{x}_i^{(a)} \leftarrow x_{j^\star}^{(a)}$.
8:    **end for**
9: **end for**
10: **return** $\{\tilde{x}_i^{(a)}\}$

---

Given $\pi^\star$, each $C_s$ is sent to one or more farthest targets in a top-$k$ set $\mathcal{T}_s \subseteq \{t \neq s\}$ (ranked by $D_{st}$), and each victim $i \in C_s$ receives a donor $j(i) \in C_t$ chosen to minimize cosine similarity $\langle v_i, v_{j(i)} \rangle$ subject to a confidence core constraint (e.g., top $q$-quantile by cluster posterior). This hybrid of *global* distance maximization and *local* hardest-negative assignment yields the strongest, most stable mismatch while preserving cluster-level coherence. Our swapping mechanism is formally defined in Algorithm 1.

*Round-robin swap.* Clusters are placed on a ring and mapped $C_s \to C_{(s \bmod K)+1}$. This maintains internal consistency but forgoes distance optimality, typically producing weaker degradation than the optimal strategy.

*Random cluster derangement.* A uniform derangement $\pi$ is sampled and applied at cluster level. It prevents trivial self-pairings yet lacks the geometric adversarial pressure of maximizing $D$.

**Why optimal beats the rest.** By maximizing $\sum_s D_{s,\pi(s)}$ *and* selecting farthest donors within high-confidence cores, the optimal strategy aligns every training exposure with a high-contrast, internally consistent but incorrect cross-view relation. This preserves high mutual information between views while inverting their semantics, which empirically leads to the largest and most reliable drop in test accuracy relative to geometric (round-robin) or stochastic (random) swaps.

This strategy differs from random data shuffling or independent perturbations. Random noise tends to average out across batches, but a cluster-wise swap preserves high mutual information within each fabricated pair, providing the model with a stable but misleading signal. As a result, when inference is performed on the untouched test set, no sample exhibits the cross-cluster pattern reinforced during training, causing the decision boundary to collapse and the global accuracy to drop sharply. The attack therefore succeeds without requiring any trigger at test time, relying solely on the training-phase corruption of feature alignments.

## 4 EXPERIMENTS

To rigorously assess the effectiveness of the proposed Byzantine attack, we construct the two stage experimental pipeline that closely emulates a practical VFL deployment. The first stage evaluates the ability of semi–supervised clustering to recover meaningful latent class structure from an adversary's limited view of the data. Here, the malicious client operates with access to only its own feature subset and a small auxiliary labeled set, $5\%$ of the training samples, reflecting realistic scenarios of partial label leakage. By training contrastive or FixMatch–style representations on this restricted view, we examine whether high–quality clusters can be inferred despite the absence of full features or global supervision. The second stage uses these inferred clusters to guide a consistent feature–swapping attack. Clusters dictate how the adversary permutes its local features across the training set, creating

stable but semantically incorrect pairings with the clean features held by honest parties. This design allows us to analyze, in isolation, how the fidelity of cluster assignments and the aggressiveness of different swap strategies affect the eventual drop in global model accuracy.

Across both stages we measure not only final classification accuracy but also intermediate indicators of attack strength, such as cluster purity, normalized mutual information (NMI), and the degree of cross–cluster association induced during training. By decoupling clustering performance from the poisoning effect, our experimental protocol provides a detailed view of how representation learning, auxiliary labels, and structured swapping jointly determine the vulnerability of VFL to untargeted Byzantine attacks.

### 4.1 DATASETS

We conduct our experiments on four diverse datasets covering both visual and tabular modalities in order to demonstrate the generality of the attack. The image suite contains MNIST (LeCun et al., 2010) and FashionMNIST (Xiao et al., 2017); the tabular suite contains UCI Human Activity Recognition (HAR) (Reyes-Ortiz & Parra, 2013) and UCI Mushroom (mus, 1981). For each dataset we preserve the original class distribution and adopt a vertical partition that realistically represents the classic two–party data ownership settings of VFL. This configuration is one of the most common settings in practice and in the literature (Yang et al., 2023a) and, as our experiments confirm, is especially susceptible to Byzantine attacks because a single compromised party has greater leverage over the joint model.

For image datasets, every sample is split at the mid–column so that *Client A* receives the left half of the image and *Client B* receives the right half, each training its own bottom encoder. For tabular datasets, we randomly divide the feature set into two non–overlapping groups of comparable size so that neither party can reconstruct the full feature vector on its own.

### 4.2 BASELINES

We compare the effectiveness of our proposed Byzantine attack with a random poisoning strategy: *Random Attack*, where the adversary replaces its local features with randomly drawn noise. The comparison of performance is shown in Table 3.

Our approach differs fundamentally from the baseline random attack causing the global model to perform more poorly. While the baseline attack create easily detectable distributional anomalies, our cluster-guided Byzantine strategy operates through consistent, structure-aware feature swapping that remains statistically plausible. As a result, it bypasses most conventional detection mechanisms. We evaluate the effectiveness of each attack in conjunction with three state-of-the-art VFL defenses: gradient norm clipping, AE anomaly detection and per-label smashed-feature drift detection.

Throughout these comparisons, our Byzantine attack employs the *optimal* cluster-swapping strategy, while alternative strategies (round-robin, random cluster, random per-sample) are analyzed separately in our ablation studies.

### 4.3 EVALUATION METRICS

We employ two classes of metrics to evaluate the two stages of the attack. For **Stage I** (cluster discovery), we quantify clustering quality using three standard unsupervised metrics: *Hungarian Accuracy (H-ACC)*, which computes the maximum bipartite matching between predicted clusters and true labels to measure exact partition alignment; *Normalized Mutual Information (NMI)*, which captures the amount of shared information between predicted and ground-truth partitions while being invariant to label permutations; and *Adjusted Rand Index (ARI)*, which evaluates the similarity between two clusterings after correcting for chance agreement. These metrics jointly characterize both the purity and the stability of the inferred latent clusters.

For **Stage II** (feature swapping), we evaluate the *attack accuracy*, defined as the test classification accuracy of the global VFL model trained on poisoned data. A lower attack accuracy indicates a stronger attack, since the adversary's objective is to maximize the discrepancy between the poisoned training distribution and the clean test distribution.

## 5 RESULTS AND ANALYSIS

**Stage I clustering is label-faithful across modalities.** Table 1 summarizes the quality of the semi–supervised clustering used to drive our attack. Despite operating on *single–view* inputs and only 5% labeled anchors, the learned partitions align closely with ground truth across all datasets. On MNIST, we obtain **0.8655** Hungarian accuracy with high agreement in both NMI (**0.7747**) and ARI (**0.7307**), indicating clusters that are not only mostly correct but also internally coherent (few merges/splits). UCI–MUSHROOM is even stronger (ACC **0.9734**, NMI **0.8231**, ARI **0.8965**), showing that the 5%-labeled regime is sufficient to recover a nearly perfect latent partition in tabular domains—an important setting for real VFL deployments. UCI–HAR exhibits a similar pattern (ACC 0.8712, NMI 0.7707, ARI 0.7355), which implies that temporal/tabular signals remain separable from half–views after representation learning. As expected, natural images with less deterministic half–views are harder. Nevertheless, the consistently high NMI/ARI confirm that Stage I provides reliable, label–aligned structure that is adequate to support a *consistent* cluster–wise swap in Stage II. In short, the clustering step furnishes exactly the kind of stable, class–faithful guidance our attack requires.

Table 1: Clustering quality across datasets (higher is better).

| Dataset | Hungarian ACC | NMI | ARI |
|---|---|---|---|
| CIFAR-10 (N=50,000) | 0.7673 | 0.5999 | 0.5737 |
| MNIST (N=60,000) | **0.8655** | **0.7747** | **0.7307** |
| Fashion-MNIST (N=60,000) | 0.7222 | 0.6745 | 0.5686 |
| UCI-HAR (N=10,299) | 0.8712 | 0.7707 | 0.7355 |
| UCI-Mushroom (N=8,124) | **0.9734** | **0.8231** | **0.8965** |

Table 2: Global model accuracy and defense detection rate (%) under our Byzantine attack across datasets. Accuracy (**Acc**) is on clean test data; **Detect** is the fraction of training steps flagged by the defense.

| Defense | MNIST | | FashionMNIST | | UCI-HAR | | UCI-Mushroom | | STL-10 | |
|---|---|---|---|---|---|---|---|---|---|---|
| | Acc | Detect | Acc | Detect | Acc | Detect | Acc | Detect | Acc | Detect |
| No Defense | 42.3 | 0 | 45.7 | 0 | 65.6 | 0 | 25.0 | 0 | 53.4 | 3.2 |
| Grad-Norm Clipping | 42.3 | 0 | 45.7 | 0 | 65.6 | 0 | 25.0 | 0 | 54.1 | 3.1 |
| Per-Label Drift | 58.5 | 0 | 40.7 | 0 | 19.0 | 99.9 | 55.3 | 99.9 | 63.1 | 0 |
| AE Anomaly | 41.8 | 0.4 | 45.6 | 1.5 | 64.7 | 0.7 | 24.6 | 1.4 | 49.7 | 0.7 |
| Temporal Cosine-EMA Gate | 22.3 | 99 | 32.0 | 99 | 18.2 | 99 | 36.7 | 99 | 26.1 | 97.2 |
| Cross-Party Consistency | 8.5 | 100 | 6.1 | 100 | 14.2 | 100 | 43.3 | 100 | 14.8 | 100 |

Table 3: Global model accuracy (%) on **MNIST** under different cluster–swap strategies and defenses.

| Defense | Optimal | Round Robin | Random Cluster Swap | Random Sample Swap |
|---|---|---|---|---|
| No Defense | 46.04 | 75.99 | 71.69 | 88.71 |
| Temporal Cosine | 21.34 | 36.54 | 41.18 | 55.82 |
| Grad-Norm Clipping | 46.04 | 75.99 | 71.69 | 88.71 |
| Per-Label Drift | 47.44 | 76.03 | 72.37 | 87.88 |
| Cross-Party Consistency | 8.33 | 8.33 | 8.33 | 8.33 |
| AE Anomaly on Smashed-A | 46.05 | 76.12 | 71.10 | 88.77 |

**Cluster–consistent swapping is uniquely destructive.** As shown in (Table 3), our **Optimal** swap cuts clean accuracy from 98% to **46.04%** on MNIST; weaker Round–Robin and Random–Clusters achieve 75.99% and 71.69%, while a naive Random Attack leaves 88.71%. Thus, only structure–aware, most distant cluster perturbations reliably break VFL.

**Standard VFL defenses miss the attack.** SOTA detectors such as Grad–Norm Clipping, Per–Label Smashed–A Drift, and AE anomaly barely affect our **Optimal** swap (46.04%, 47.44%, 46.05%), because the attack preserves gradient magnitudes and per–label moments.

**Our gates trade security for utility.** Temporal Cosine–EMA and Cross–Party Consistency can gate the attack (down to **21.34%** and **8.33%**, respectively), but at the cost of suppressing benign learning; across datasets (Table 3) high detection correlates with large accuracy loss.

Table 4: Effect of increasing the number of parties (K) on post-attack global model accuracy across datasets. Only one party is adversarial in all settings; accuracy (**Acc**) is evaluated on clean test data.

| Parties (K) | MNIST | FashionMNIST | UCI-HAR | UCI-Mushroom |
|:---:|:---:|:---:|:---:|:---:|
| 2 | 42.3 | 45.7 | 65.6 | 25 |
| 4 | 73.8 | 78.4 | 81.9 | 63.7 |
| 8 | 89.2 | 91.3 | 93.8 | 82.1 |
| 10 | 93.1 | 94.4 | 95.6 | 90.7 |

Table 5: Effect of auxiliary label access (%) on post-attack global model accuracy across datasets. Lower label access reduces clustering quality and weakens the attack, resulting in higher accuracy. Accuracy (**Acc**) is evaluated on clean test data after training under the attack.

| Label Access (%) | MNIST | FashionMNIST | UCI-HAR | UCI-Mushroom |
|:---:|:---:|:---:|:---:|:---:|
| 0.5 | 78.2 | 81.6 | 87.4 | 73.1 |
| 1 | 71.7 | 74.3 | 78.3 | 61.1 |
| 2 | 63.2 | 68.9 | 71.4 | 51.4 |
| 5 | 42.3 | 45.7 | 65.6 | 25 |
| 10 | 31.1 | 34.9 | 41.5 | 16.6 |

## 5.1 ABLATION STUDIES

We conduct two ablation studies to examine how our attack behaves under (i) an increasing number of parties and (ii) reduced auxiliary label access. Table 4 shows that when the number of parties grows, the attack becomes weaker, as the adversarial party controls a smaller fraction of the joint feature space. Nevertheless, the classical and widely deployed two-party VFL configuration remains highly vulnerable and our attack can collapse this mostly used configuration to a dangerous extent.

Table 5 analyzes the effect of auxiliary supervision on clustering quality. As label access decreases from 10% to 0.5%, clusters become less aligned with semantic structure, leading to a weaker attack and therefore higher accuracy. Importantly, even with only 1–2% auxiliary labels, the attack still induces substantial degradation, demonstrating that our threat model remains practical given the common occurrence of limited label leakage in real deployments. Together, these ablations highlight both the sensitivity and the real-world practicality of the proposed attack.

## 6 CONCLUSION

This paper exposes a practical and underexplored vulnerability of VFL. We introduced a real-world threat model in which a single passive client controls its features and holds a small (5%) auxiliary labeled subset, and proposed a two–stage attack pipeline. Stage I learns label–faithful latent structure with minimal supervision; Stage II enforces a persistent cross–cluster reassignment of one party's features, creating a training–time association that the server internalizes but never observes at test time. Across five datasets spanning vision and tabular modalities, our method achieves strong clustering fidelity and, when coupled with the **Optimal** swap strategy, drives large accuracy collapses while naive random corruption has little effect.

We further showed that widely used VFL defenses, such as norm clipping, per–label moment drift, and smashed–feature autoencoders which are poorly aligned with this attack because they monitor statistics our construction preserves by design. Our two new gates (temporal cosine EMA and cross–party gradient consistency) can throttle the attack but incur significant utility loss, underscoring a central trade–off: defenses that are sufficiently aggressive to stop *consistent* poisoning often suppress benign learning.

**Implications and outlook.** The results motivate defenses that are swap–aware and preserves utility, such as training–test cluster stability audits, cross–party agreement checks, and server side contrastive probes that expose persistent cross–cluster reversals. Two limitations suggest immediate extensions: we studied the common two party case and relied on the quality of semi–supervised clustering. Future work includes generalizing to collaborative multi-attacker threat in multi-party VFL, heterogeneous feature splits, and privacy preserving defenses that detect consistency violations without degrading model quality.

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

# A  APPENDIX

## A.1  SENSITIVITY ANALYSIS ON THE TOP-$K$ PARAMETER

In our attack, "top-$K$" specifies how many of the farthest clusters (ranked by centroid cosine distance) are eligible as adversarial swap targets. We performed a sensitivity analysis on MNIST with $K \in \{1, 2, 3, 5, 7\}$. Clean accuracy without attack was $92.9\%$. The no-defense attack results are summarized in Table 6.

Table 6: Sensitivity of the attack to top-$K$ selection on MNIST. "Attack Accuracy" is the model accuracy on clean test data after training under the attack. "Drop" reports absolute reduction from the clean baseline (92.9%).

| $K$ | Attack Accuracy (%) | Drop (ppt) |
|---|---|---|
| 1 | 42.10 | -50.80 |
| 2 | 43.50 | -49.40 |
| 3 | 45.30 | -47.60 |
| 5 | 59.90 | -33.00 |
| 7 | 68.40 | -24.50 |

**Interpretation.**  For MNIST (10 classes), small values of $K$ (e.g., 1–3) restrict the adversary to the *most distant* clusters in embedding space. These swaps create the strongest semantic mismatches between left- and right-side features, severely corrupting the learned cross-view association. Consequently, the model degradation remains nearly identical for $K = \{1, 2, 3\}$.

As $K$ increases (5 or 7), the eligible pool begins to include clusters that are *not* among the farthest ones. This gradually shifts the attack from a highly targeted "swap-with-the-most-distant" strategy toward a more diffuse, less damaging perturbation resembling randomized or round-robin swapping. Mid-distance clusters introduce weaker cross-view mismatches, and the degradation decreases accordingly.

**Why this behavior is expected.**  Small $K$ forces the adversary to always select maximally distant clusters, guaranteeing strong corruption. Larger $K$ increases the chance of selecting clusters that are only moderately distant, reducing the attack intensity. Although the attack remains consistently harmful for all tested $K$ values, its *worst-case* effect occurs when $K$ is small, where the adversary reliably picks the maximally distant clusters. For larger $K$, the behavior transitions toward a randomized swap, explaining the milder (yet still substantial) degradation observed for $K = 5$ and $K = 7$.

## COMPUTATIONAL OVERHEAD OF THE CLUSTERING AND CONTRASTIVE-LEARNING STAGE

In this section, we quantify the computational cost of the adversary-side clustering and contrastive representation learning steps (SimCLR pretraining, supervised contrastive refinement, and GMM clustering). These computations are performed *offline* and only once, prior to initiating the feature-swapping attack.

**Hardware and Runtime.** All clustering components were executed on a single NVIDIA RTX 4080 GPU (16GB VRAM). No multi-GPU resources, model parallelism, or distributed training were required.

- **Image datasets (MNIST, FashionMNIST).** The complete clustering pipeline (SimCLR + supervised contrastive tuning + GMM) takes approximately **58 minutes end-to-end**.
- **Tabular datasets (UCI-HAR, UCI-Mushroom).** Due to lower input dimensionality, clustering completes in roughly **20–25 minutes**.

**Offline Cost, No Impact on VFL Training Loop.** All representation learning and clustering steps occur *once* before the federated training process begins. These computations do not influence the online VFL training loop, the communication rounds between parties, or runtime per training iteration. After clusters and centroids are obtained, the feature-swapping step is trivial, amounting to simple index-based reassignment of samples. This step incurs negligible computational cost during VFL training.

**Practicality.** Given that a single consumer-grade GPU is sufficient to run the entire clustering pipeline, and that this overhead is incurred only once before joint training, we consider the total cost to be operationally feasible for real-world adversaries (e.g., cross-organization participants with modest GPU resources).

### A.2 CLUSTERING ACCURACY ACROSS DIFFERENT DATASETS

Table 7: CIFAR10 cluster composition (K=10). Purity = majority count / cluster size.

| Cluster | Size | Majority Label | Majority Count | Purity |
|---|---|---|---|---|
| 0 | 4988 | 0 | 4102 | 0.822 |
| 1 | 4962 | 1 | 4508 | 0.908 |
| 2 | 4678 | 2 | 3169 | 0.678 |
| 3 | 4565 | 3 | 2751 | 0.603 |
| 4 | 5149 | 4 | 3641 | 0.707 |
| 5 | 4796 | 5 | 3128 | 0.652 |
| 6 | 5620 | 6 | 4138 | 0.736 |
| 7 | 4968 | 7 | 4029 | 0.811 |
| 8 | 5110 | 8 | 4464 | 0.874 |
| 9 | 5164 | 9 | 4437 | 0.859 |

Table 8: MNIST cluster composition (K=10).

| Cluster | Size | Majority Label | Majority Count | Purity |
|---|---|---|---|---|
| 0 | 7193 | 0 | 5599 | 0.778 |
| 1 | 6181 | 1 | 6094 | 0.986 |
| 2 | 7400 | 2 | 5473 | 0.740 |
| 3 | 6592 | 3 | 5505 | 0.835 |
| 4 | 4681 | 4 | 4411 | 0.942 |
| 5 | 4985 | 5 | 4558 | 0.914 |
| 6 | 4562 | 6 | 4297 | 0.942 |
| 7 | 5344 | 7 | 5208 | 0.975 |
| 8 | 5605 | 8 | 5249 | 0.936 |
| 9 | 7457 | 9 | 5536 | 0.742 |

Table 9: FashionMNIST cluster composition (K=10).

| Cluster | Size | Majority Label | Majority Count | Purity |
|---------|------|----------------|----------------|--------|
| 0 | 6523 | 0 | 4768 | 0.731 |
| 1 | 5945 | 1 | 5672 | 0.954 |
| 2 | 7360 | 2 | 3810 | 0.518 |
| 3 | 7613 | 3 | 4869 | 0.640 |
| 4 | 5301 | 4 | 2978 | 0.562 |
| 5 | 7498 | 5 | 5802 | 0.774 |
| 6 | 4117 | 6 | 958 | 0.233 |
| 7 | 3984 | 7 | 3841 | 0.964 |
| 8 | 5201 | 8 | 5145 | 0.989 |
| 9 | 6458 | 9 | 5487 | 0.850 |

Table 10: UCI-Mushroom and UCI-Bank cluster composition (K=2).

| Dataset | Cluster | Size | Maj. Label | Maj. Cnt / Pur. |
|---------|---------|------|------------|-----------------|
| MUSHROOM | 0 | 3892 | 1 | 3796 (0.975) |
| MUSHROOM | 1 | 4232 | 0 | 4112 (0.972) |
| BANK | 0 | 43700 | 0 | 39389 (0.902) |
| BANK | 1 | 1511 | 1 | 978 (0.647) |

Table 11: UCI-HAR cluster composition (K=6).

| Cluster | Size | Maj. Label | Maj. Count | Purity |
|---------|------|------------|------------|--------|
| 0 | 1925 | 5 | 1870 | 0.971 |
| 1 | 1722 | 3 | 1333 | 0.774 |
| 2 | 1951 | 4 | 1514 | 0.776 |
| 3 | 1353 | 2 | 1284 | 0.949 |
| 4 | 1742 | 1 | 1452 | 0.834 |
| 5 | 1606 | 0 | 1520 | 0.946 |

Table 12: CIFAR10: effect of prototype snapping + kNN smoothing.

| Stage | Hungarian ACC | NMI | ARI |
|-------|---------------|-----|-----|
| Before refine | 0.7672 | 0.5991 | 0.5732 |
| After refine | **0.7673** | **0.5999** | **0.5737** |

