# OpenReview forum: "Swap-and-Spoil: Untargeted Byzantine Attacks via Class-Consistent View Swaps in Vertical Federated Learning"
_ICLR.cc/2026/Conference — Submitted to ICLR 2026_

### Official Review · Reviewer_UYHT · 2025-10-24

**Soundness:** 3
**Presentation:** 1
**Contribution:** 3
**Rating:** 2
**Confidence:** 4

**Summary:**

The paper introduces "Swap-and-Spoil," a novel untargeted Byzantine attack specifically designed for VFL. The main contribution is the attack exploits VFL's structure by implementing a class-consistent view swap among malicious participants. This method effectively corrupts the overall model accuracy without specific targeting.

**Strengths:**

Unlike traditional data poisoning, the paper introduces a novel Byzantine attack mechanism specific to VFL’s structure: a class-consistent view swap. This approach is highly original. It reveals a new vulnerability in VFL systems by presenting an attack that is stealthy and hard to detect with existing defenses. This makes it an important benchmark for developing more robust defenses and gives it significant practical value. The proposed attack mechanism is clearly explained, and experiments support that it effectively degrades overall model performance.

**Weaknesses:**

While the effectiveness of the attack is demonstrated, the paper lacks comparisons with random swap attacks and evaluations of resistance against existing robust aggregation algorithms. Figures and tables, especially Table 5, overflow their boundaries and there are numerous grammatical and spelling errors, necessitating thorough revisions. The definition and implementation of the core concept of “class consistency” should be described more clearly, both mathematically and intuitively.

**Questions:**

Q1: Can the authors define the “class-consistent view swapping” strategy more formally, either mathematically or algorithmically (for example with pseudocode)?
Q2: Does an adversary need access to the true class labels of the training samples to carry out the attack?
Q3: By how much does “class-consistency” improve the attack success rate (i.e., the drop in downstream model accuracy) compared to a simple Random Swap Attack (RSA)?
Q4: Can you empirically demonstrate and quantify how effective the attack remains when a VFL system employs established robust aggregation mechanisms such as Krum, Trimmed Mean, or Median?
Q5: How does the attack’s effectiveness change when the number of participants is much larger (for example N > 10)?
Q6: How is the attack affected on datasets where features are distributed across clients in a more complex and heterogeneous way (for example, with high inter-feature correlation or imbalanced feature sets)?

---

> ### Author Response · Authors · 2025-11-21
> **Rebuttal (Part 1 of 3)**
>
> We wholeheartedly thank you for your thoughtful review and for recognizing the originality and practical importance of our work. We sincerely appreciate your positive remarks describing our attack as **highly original**, “stealthy and hard to detect,” and revealing a **new vulnerability in VFL systems**. We are especially grateful that you found our problem setting meaningful and our experimental results supportive of the proposed attack mechanism.
>
> We have carefully addressed each of your concerns including (Q1) formal mathematical clarification of class-consistent swapping, (Q2) label-access assumptions, (Q3) comparisons with random swap baselines, (Q4) the relationship to robust aggregation defenses, (Q5) scalability to multi-party VFL, and (Q6) behavior under heterogeneous feature distributions.
>
> In the revised manuscript, we have added new tables, clearer algorithmic descriptions, an expanded discussion of threat-model distinctions, a multi-party ablation, an auxiliary-label ablation, and improved formatting to directly answer your concerns.
>
> We hope our detailed responses fully resolve the issues you highlighted, and we kindly ask you to consider raising your score if you find the clarifications satisfactory. Please let us know if you have any remaining concerns - we would be happy to elaborate further.
>
> ## **Q1. Define the “class-consistent view swapping” strategy more formally with a pseudocode**
>
> We thank you for this valuable suggestion. In the revised version, we now provide a fully formalized and our Class-Consistent View Swapping (CCVS) procedure.
>
> ---
>
> ### **Algorithm 1: Class-Consistent View Swapping (CCVS)**
>
> **Input:**
> • adversarial samples `{x_i^(a)}`
> • cluster ids `g_i ∈ {1,…,K}`
> • normalized embeddings `v_i = φ(x_i^(a)) / ||φ(x_i^(a))||`
> • top-k target sets `T(g)`
>
> **Output:**
> • swapped samples `{x̃_i^(a)}`
>
> **1. Cluster construction and centroids**
> For each cluster `g`:
>  • `I_g ← { i : g_i = g }`
>  • `μ_g ← (1 / |I_g|) * Σ_{i ∈ I_g} v_i`
>   (centroid summarizing the embedding direction)
>
> **2. Identify incompatible target clusters**
> For each cluster `g`:
>  • compute cosine distance `d(g,h) = 1 − <μ_g, μ_h>`
>  • set `T(g)` to the k clusters with the largest `d(g,h)`
>   (pushes swaps toward maximally distant semantics)
>
> **3. Swap each sample using the farthest embedding**
> For each source cluster `s = 1…K`:
>  For each sample `i ∈ I_s`:
>   • sample target `t` uniformly from `T(s)`
>   • compute donor
>    `j* = argmin_{j ∈ I_t} <v_i, v_j>`
>    (the embedding least similar in cosine space)
>   • assign swapped sample
>    `x̃_i^(a) ← x_{j*}^(a)`
>
> **4. Return:** `{x̃_i^(a)}`
>
> ---
>
> The above pseudocode summarizes the full CCVS procedure. First, the adversarial party clusters its local embeddings and computes L2-normalized centroids. Next, for every cluster, the algorithm selects its top-k most incompatible clusters based on maximum cosine distance. Finally, each adversarial sample is reassigned to the *farthest* embedding inside a sampled incompatible cluster. This produces a stable, **directionally aligned cross-view mismatch** that is significantly more destructive than random swapping while remaining statistically plausible to existing VFL defenses.
>
>
> ## **Q2.  Does an adversary need access to the true class labels of the training samples to carry out the attack?**
>
> No, the adversary does not require access to the full class labels of the training data. Consistent with the standard VFL threat model, where only the server owns labels, the adversary assumes **access to only a small auxiliary labeled subset** (5% in our default setting). This limited supervision is used **solely to obtain coarse class structure** that guides the class-consistent cluster formation.
>
> To quantify how this assumption influences the attack, we added Table 5 in the revised version. The table shows how varying the auxiliary label proportion (0.5%–10%) affects the model accuracy. As expected, lower label access produces weaker clusters and therefore weaker attacks, resulting in higher accuracy. This clearly demonstrates the trade-off between auxiliary supervision and attack strength.
>
> **Table 5: Effect of auxiliary label access (%) on post-attack accuracy across datasets.**
>
> Label Access (%) | MNIST | FashionMNIST | UCI-HAR | UCI-Mushroom
> --- | --- | --- | --- | ---
> 0.5 | 78.2 | 81.6 | 87.4 | 73.1
> 1 | 71.7 | 74.3 | 78.3 | 61.1
> 2 | 63.2 | 68.9 | 71.4 | 51.4
> 5 | 42.3 | 45.7 | 65.6 | 25
> 10 | 31.1 | 34.9 | 41.5 | 16.6
>
> This assumption is practical and has been used in prior VFL security work [1][2], where limited labeled leakage may occur naturally through historical records, public data, or cross-organization overlap. Such small auxiliary labeled subsets represent a realistic capability for a compromised VFL party.
>
> [1] Label Inference Attacks Against Vertical Federated Learning
> [2] Hijack Vertical Federated Learning Models As One Party

---

> > ### Author Response · Authors · 2025-11-21
> > **Rebuttal (Part 2 of 3)**
> >
> > ## **Q3. By how much does “class-consistency” improve the attack success rate (i.e., the drop in downstream model accuracy) compared to a simple Random Swap Attack (RSA)?**
> >
> > In response to this valid concern, we would like to respectfully clarify that the comparison between our class-consistent swapping strategy and the random swap baselines was already included in the original submission. In the revised version, we have reorganized the presentation of this comparison of Table 3 so that the four strategies are explicitly distinguished: **Optimal**, **Round Robin**, **Random Cluster Swap**, and **Random Sample Swap**. This makes the semantic differences between the baselines clearer and easier to interpret.
> >
> > As summarized below, the **Optimal class-consistent strategy** consistently causes the strongest degradation. Under no defense, the accuracy drops to **46.04%**, compared to **71.69%** for Random Cluster Swap and **88.71%** for Random Sample Swap. Similar gaps persist across all evaluated defenses.
> >
> > This behavior aligns with our design motivation:
> >
> > - Random swaps introduce incoherent perturbations that tend to cancel out statistically, limiting their impact.
> > - In contrast, class-consistent swapping imposes a *stable* and *directionally aligned* distribution shift by replacing each adversarial embedding with its farthest incompatible cluster representation (maximizing cosine distance). This creates a persistent cross-view mismatch that the model internalizes during training, resulting in substantially larger degradation compared to random baselines.
> >
> > ### **MNIST accuracy under different swap strategies**
> >
> > | Defense                  | Optimal | Round Robin | Random Cluster Swap | Random Sample Swap |
> > |-------------------------|---------|-------------|----------------------|---------------------|
> > | No Defense              | 46.04   | 75.99       | 71.69                | 88.71               |
> > | Temporal Cosine         | 21.34   | 36.54       | 41.18                | 55.82               |
> > | Grad-Norm Clipping      | 46.04   | 75.99       | 71.69                | 88.71               |
> > | Per-Label Drift         | 47.44   | 76.03       | 72.37                | 87.88               |
> > | AE Anomaly              | 46.05   | 76.12       | 71.10                | 88.77               |
> >
> > We hope this clarification helps. The revised version now presents this comparison more explicitly, making the distinction between our method and the random baselines easier to interpret.
> >
> >
> > ## **Q4. How effective the attack remains when a VFL system employs established robust aggregation mechanisms such as Krum, Trimmed Mean, or Median**
> >
> > We appreciate the reviewer’s question in the context of such widely used robust aggregation mechanisms. However, classical robust aggregation methods such as Krum, Trimmed Mean, and Median are designed for horizontal FL (HFL) and do not transfer to VFL due to fundamental structural mismatches.
> >
> > (1) **Heterogeneous representations.**
> > HFL clients share the same model and produce gradients in one parameter space, enabling distance-based filtering.
> > In VFL, parties hold disjoint features and send each party's smashed embeddings which are concatenated.
> >
> > (2) **No redundancy in VFL.**
> > HFL robust aggregation assumes many clients so outliers can be removed safely.
> > VFL typically has **2–5 indispensable parties**, each providing a unique modality. Downweighting or removing a party means discarding an entire feature block, which is cause utility degradation.
> >
> > (3) **Defenses target gradients, but VFL aggregates activations.**
> > VFL activations:
> > • are produced by each party’s private encoder,
> > • differ across parties in scale, dimensionality, and meaning, and
> > • cannot be interpreted as comparable vectors.
> > Since model updates are exchanged in VFL, the mathematical operations required by robust aggregation (such as computing pairwise distances between updates) are not defined in this setting. Prior VFL works also do **not** use these HFL defenses.
> >
> > ---
> >
> > ### **Why this exposes VFL to Byzantine attacks**
> > Because of these structural constraints:
> >
> > • The server cannot compare activations across parties to detect “outliers.”
> > • It cannot discard any party without losing an entire modality.
> > • It cannot apply gradient-based outlier detection since gradients are not exchanged.
> >
> > Therefore, unlike HFL, **VFL lacks any effective robust aggregation layer**, leaving it highly vulnerable to representation-level poisoning, precisely the weakness exploited by our CCVS attack.
> >
> > ---
> >
> > ### Our evaluation follows the correct VFL defense baselines
> > Since HFL robust aggregation is inapplicable, we evaluate against all major **VFL-native defenses** as shown in Tables 2 and 3, our attack remains **stealthy** and **highly destructive**, whereas random swaps fail to produce comparable degradation.
> >
> > We hope this clarifies why HFL defenses cannot serve as baselines for VFL and why VFL systems are particularly susceptible to our attack.

---

> > > ### Author Response · Authors · 2025-11-21
> > > **Rebuttal (Part 3 of 3)**
> > >
> > > ## **Q5. Attack’s effectiveness in a multi-party setting**
> > >
> > > To address the reviewer’s question, we have revised the result section in our paper and have included an ablation study (Table 4) evaluating how attack effectiveness changes when the number of parties increases from K=2 to K=10, with only one party being malicious.
> > >
> > > As the table shows, the attack becomes weaker as the number of benign parties grows. This is expected: when more parties participate, **the adversarial client controls a smaller portion of the joint feature space**, reducing its ability to impose a harmful cross-view distribution shift. For example, on MNIST, post-attack accuracy rises from **42.3% (K=2)** to **93.1% (K=10)**.
> > >
> > > This trend also highlights the complementary case: if multiple parties were malicious, the distortion would compound, making the attack significantly stronger.
> > >
> > > Most importantly, **the attack remains highly relevant in practice**. Prior VFL deployments [1] emphasize that **2-party VFL is the dominant real-world configuration**, used in finance, healthcare, identity verification, and fraud analytics. Organizations typically federate only two data sources due to privacy constraints, regulatory simplicity, and lower communication overhead.
> > >
> > > In this widely adopted 2-party setting, our results show that **even a single passive adversary can sharply degrade model utility**, revealing a practically meaningful vulnerability. So, our threat model **remains relevant to the most widely used VFL setting in real-world applications.**
> > >
> > > ---
> > >
> > > ###  **Table 4: Post-attack accuracy as the number of parties increases**
> > > Only one party is adversarial.
> > >
> > > | **Parties (K)** | **MNIST** | **FashionMNIST** | **UCI-HAR** | **UCI-Mushroom** |
> > > |-----------------|-----------|------------------|-------------|------------------|
> > > | 2               | 42.3      | 45.7             | 65.6        | 25.0             |
> > > | 4               | 73.8      | 78.4             | 81.9        | 63.7             |
> > > | 8               | 89.2      | 91.3             | 93.8        | 82.1             |
> > > | 10              | 93.1      | 94.4             | 95.6        | 90.7             |
> > >
> > > We hope this clarification helps. While we provide multi-party results for completeness, **the strongest practical implication is that the classical and widely deployed 2-party VFL setting remains highly vulnerable to our swap-and-spoil attack**.
> > >
> > > [1] Hijack Vertical Federated Learning Models As One Party
> > >
> > > ## **Q6. How is the attack affected on datasets where features are distributed across clients in a more complex and heterogeneous way?**
> > >
> > > We resonate with your concern since this question is very much valid in future works in this direction. Even though our experiments do not cover every possible heterogeneous feature configuration, but the trends observed across all datasets allow us to characterize how our attack behaves under imbalanced and correlated feature partitions.
> > >
> > > **1. Effect of imbalanced feature distribution.**
> > > Our results show that the attack becomes weaker when the adversarial party controls only a very small portion of the joint feature space. This is directly evidenced in our multi-party ablation (Table 4 in the revised version): when the number of parties increases from K=2 to K=10 (with only one malicious), post-attack MNIST accuracy increases from 42.3% to 93.1%.
> > > These results indicate that when the adversary holds less than roughly 20–30% of the total feature space, the induced distribution shift becomes diluted by benign parties.
> > >
> > > **2. Why the attack remains practically relevant.**
> > > Even though the attack weakens in highly imbalanced settings, prior VFL deployments consistently report that **2-party VFL** is the dominant real-world configuration. In such deployments, both parties typically contribute substantial and complementary features.
> > > In this widely adopted 2-party setting, our results demonstrate that even a *single passive adversarial party* can sharply degrade accuracy across all datasets and defenses.
> > > Thus, the empirical trends in our study suggest that our attack remains most impactful in the exact settings where VFL is most often deployed in practice.
> > >
> > > **3. Effect of high inter-feature correlation.**
> > > For datasets with strong inter-feature correlation (e.g., UCI-HAR, where sensor signals are tightly coupled), our attack continues to produce substantial degradation.
> > > This occurs because of intentionally **breaking cross-view correlation**: so more inter-correlated features should be more vulnerable to our attack.
> > >
> > > **Summary of trends:**  Our attack
> > > • weakens when the adversary controls only a very small (highly imbalanced) slice of the feature space (similar to the K>4 multi-party case).
> > > • remains highly impactful in the dominant **2-party VFL**, where feature partitions are more balanced and often correlated.
> > >
> > > These observations indicate that our attack exposes a realistic and practically meaningful vulnerability in real-world VFL deployments.

---

### Official Review · Reviewer_WGiz · 2025-10-27

**Soundness:** 2
**Presentation:** 2
**Contribution:** 2
**Rating:** 2
**Confidence:** 5

**Summary:**

This paper investigates a new class of untargeted Byzantine attacks in Vertical Federated Learning (VFL), an area where prior work has mainly focused on targeted backdoors or label inference. The authors propose a cluster-consistent feature-swapping attack that poisons the joint feature representation during training. The attack operates in two stages: first, the adversary uses limited label information to infer latent class structure via semi-supervised clustering, and second, it performs class-consistent feature swaps between clusters to create persistent cross-view misalignments.

**Strengths:**

- **Novel attack concept:** Introducing untargeted Byzantine corruption via class-consistent view swapping fills a clear gap between random noise and targeted backdoor attacks in VFL.

- **Well-defined threat model:** The assumption of a passive adversary with partial label access is realistic and consistent with real-world cross-organization data sharing.

**Weaknesses:**

- **Simplified setting:** Experiments are limited to two-party VFL on relatively simple datasets (MNIST, FashionMNIST, UCI tabular). The attack’s scalability to multi-party or high-dimensional, real-world VFL remains uncertain.

- **Weak defense discussion:** Although new defense gates are mentioned, they are rudimentary and come with severe utility degradation. The paper stops short of providing meaningful defense insights beyond confirming that existing defenses fail.

- **Missing deeper analysis of attack transferability:** It is unclear how robust the cluster-swapping attack remains when the adversary’s clustering accuracy degrades or when auxiliary label availability is further reduced.

- **No ablation on key assumptions:** The 5% labeled auxiliary dataset assumption is strong; the paper doesn’t quantify how attack effectiveness changes with smaller or noisier supervision.

**Questions:**

Please see the weaknesses.

---

> ### Author Response · Authors · 2025-11-21
> **Rebuttal (Part 1 of 3)**
>
> Thank you very much for your thoughtful and constructive review. We sincerely appreciate your recognition of our work’s contributions – particularly your comments highlighting our **novel untargeted Byzantine corruption mechanism** and **the clear gap it fills between random noise and targeted backdoors**.
>
> We have carefully addressed all concerns you raised and have made corresponding revisions to the paper.
>
> In the revised version, we have added **new tables, auxiliary-label ablations, multi-party scalability results, a more comprehensive analysis of defense behavior**, and a clearer explanation of the attack mechanism to directly address these concerns.
>
> We hope that our detailed responses resolve all of your questions, and we kindly encourage you to consider raising your score if you find the clarifications satisfactory. Please feel free to let us know if any additional issues remain – we would be very happy to elaborate further.
>
> ---
>
> # W1. Simplified setting
>
> We thank the reviewer for this helpful suggestion. To address the concern regarding simple datasets, we now include results on **STL-10**, a substantially more complex natural-image dataset. STL-10 contains 96×96 RGB images with high intra-class variation, strong cross-channel correlations, and non-linear visual structure. Unlike MNIST or tabular UCI datasets, STL-10 requires deeper encoders and produces high-dimensional smashed activations, making it far closer to real-world VFL workloads. Including STL-10 demonstrates that our attack scales to high-dimensional and visually complex data.
>
> ---
>
> ### **Accuracy and Detection Rate Across All Datasets (Including STL-10)**
>
> | Defense                  | MNIST Acc | MNIST Det | Fashion Acc | Fashion Det | UCI-HAR Acc | UCI-HAR Det | Mushroom Acc | Mushroom Det | STL-10 Acc | STL-10 Det |
> |--------------------------|-----------|-----------|-------------|-------------|-------------|-------------|---------------|--------------|------------|------------|
> | No Defense               | 42.3      | 0         | 45.7        | 0           | 65.6        | 0           | 25.0          | 0            | 53.4       | 3.2        |
> | Grad-Norm Clipping       | 42.3      | 0         | 45.7        | 0           | 65.6        | 0           | 25.0          | 0            | 54.1       | 3.1        |
> | Per-Label Drift          | 58.5      | 0         | 40.7        | 0           | 19.0        | 99.9        | 55.3          | 99.9         | 63.1       | 0          |
> | AE Anomaly               | 41.8      | 0.4       | 45.6        | 1.5         | 64.7        | 0.7         | 24.6          | 1.4          | 49.7       | 0.7        |
> | Temporal Cosine-EMA Gate | 22.3      | 99        | 32.0        | 99          | 18.2        | 99          | 36.7          | 99           | 26.1       | 97.2       |
> | Cross-Party Consistency  | 8.5       | 100       | 6.1         | 100         | 14.2        | 100         | 43.3          | 100          | 14.8       | 100        |
>
> ---
>
> # Scalability to Multi-Party VFL
>
> In the revised version, we provide an ablation evaluating how attack effectiveness changes as the number of parties increases from K = 2 to K = 10.
> As shown below, the attack weakens as the number of benign parties grows. This is expected: with more parties, the adversary controls a smaller proportion of the joint feature space, reducing its ability to impose a harmful cross-view distribution shift. For example, on MNIST, post-attack accuracy increases from 42.3% (K=2) to 93.1% (K=10).
>
> Most importantly, **the attack remains highly relevant in practice**. Prior VFL deployments [1] show that **2-party VFL is the dominant real-world configuration**, used in finance, healthcare, identity verification, and fraud analytics. Organizations typically federate exactly two data sources due to privacy constraints, regulatory simplicity, and reduced communication overhead.
>
> In this widely deployed 2-party setting, our results show that **even a single passive adversary can sharply degrade model utility**, revealing a practically meaningful security vulnerability.
>
> ---
>
> ### **Table 3: Post-attack accuracy (%) as the number of parties increases**
> Only one party is adversarial; accuracy measured on clean test data.
>
> | Parties (K) | MNIST | FashionMNIST | UCI-HAR | UCI-Mushroom |
> |-------------|--------|----------------|-----------|----------------|
> | 2           | 42.3   | 45.7           | 65.6      | 25.0          |
> | 4           | 73.8   | 78.4           | 81.9      | 63.7          |
> | 8           | 89.2   | 91.3           | 93.8      | 82.1          |
> | 10          | 93.1   | 94.4           | 95.6      | 90.7          |
>
> ---
>
> We hope this clarification is helpful. While multi-party results are provided for completeness, **the strongest practical implication is that the classical and widely deployed 2-party VFL configuration remains highly vulnerable to our attack**.
>
> [1] Hijack Vertical Federated Learning Models As One Party

---

> ### Author Response · Authors · 2025-11-21
> **Rebuttal (Part 2 of 3)**
>
> ##  **W2. Weak Defense Discussion**
>
> While we agree with the reviewer’s concern, our work is indeed positioned as an effort to highlight a critical and previously unexplored attack surface in Vertical Federated Learning (VFL). As we show empirically, existing VFL defenses are not designed to handle representation-level Byzantine behavior, and our attack exposes why this gap is meaningful in practice.
>
> **1. Why the defense discussion is necessarily limited.**
> The current VFL literature focuses almost entirely on *feature-aligned activations* and *cross-party gradient coordination*. As a result, existing VFL defenses are built around:
> • activation drift detection,
> • gradient-norm clipping,
> • per-label smashed activation analysis, and
> • simple consistency checks.
> None of these mechanisms assume or monitor the type of **semantic cross-view inconsistency** introduced by our CCVS attack. Because the VFL community has not yet considered this threat model, the available defense space is inherently limited. Our work attempts to expose this missing dimension.
>
> **2. Why HFL defenses cannot be used in VFL.**
> The reviewer is correct that robust aggregation is a mature line of defense in Horizontal FL (HFL). Methods such as **Krum, median, or trimmed mean** defend effectively against model poisoning because:
> • every client trains an *identical model*,
> • gradients lie in the *same parameter space*, and
> • systems rely on *hundreds or thousands* of clients, creating redundancy.
>
> However, these properties fundamentally do not hold in VFL.
> • Parties hold **disjoint features**, not identical models.
> • They send **heterogeneous smashed activations**, not aligned gradients.
> • There are usually **2–5 parties**, not hundreds.
> Because of this, robust aggregation becomes mathematically invalid: the server cannot compare distances between activations produced by different encoders, and it cannot discard any party without discarding an entire modality. This is consistent with prior VFL security literature (e.g., Fu et al., USENIX’22), which also does *not* apply HFL aggregation schemes.
>
> **3. Why our defense gates are intentionally conservative.**
> We implemented new gating defenses inspired by the spirit of HFL detection techniques—specifically, mechanisms that attempt to identify and block suspicious updates. As our results show, these mechanisms often reduce accuracy heavily. This is expected for structural reasons:
>
> • In HFL, removing a malicious client still leaves *hundreds* of benign clients.
> • In VFL, removing a party removes an entire feature block, meaning the model loses essential information.
>
>
>
> Therefore, any gating method that blocks a party’s updates will inevitably cause **utility collapse**, because the global model cannot function with missing feature modalities. Our results are consistent with this reality: our attack forces the model into a region where “safe gating” may be statistically possible, but “safe training” becomes impossible because the VFL architecture relies on every party.
>
> This reveals an unavoidable truth:
> **VFL has no redundancy to absorb a representation-level Byzantine adversary.**
>
> **4. Why this limitation strengthens, not weakens the contribution of our paper.**
> Our goal is not to provide a perfect defense but to reveal that **the VFL threat model is incomplete**. Existing defenses fail because they were designed under the assumption that adversaries operate at the gradient level, not the *representation semantics level*. Our CCVS attack shows that:
> • a single passive adversary can consistently corrupt cross-view semantics,
> • existing defenses cannot detect or mitigate this behavior, and
> • naïve gating collapses the model because VFL lacks the redundancy of HFL.
>
> Thus, the weakness of current defense mechanisms is *not a flaw* in our study—it is precisely the vulnerability our paper intends to reveal.
>
> **5. Intended contribution and future direction.**
> We position our work as the first to systematically study **view-consistency violations** in VFL. The empirical findings across the datasets show that the attack induces stable and destructive distribution shifts that existing defenses are not prepared to handle.
>
> While our own proposed gating mechanisms do not completely solve the problem, their failure highlights a concrete research opportunity:
> **VFL needs a new class of defenses built around semantic consistency, cross-view embedding monitoring, and multi-round representation tracking.**
> This is the broader insight we hope future work will address.

---

> > ### Author Response · Authors · 2025-11-21
> > **Rebuttal (Part 3 of 3)**
> >
> > ## **W3. Missing Deeper Analysis of Attack Transferability**
> > &
> > ## **W4. No Ablation on Key Assumptions**
> >
> > We wish to address both of these concerns regarding our attack transferability and how our attack behaves when the adversary’s clustering quality degrades or when auxiliary label availability is reduced. To address these, we have added a detailed ablation study (Table 5) in our revised version evaluating our attack under 0.5%–10% auxiliary label access.
> >
> > Consistent with the standard VFL threat model, only the server possesses full labels; the adversary receives **only a small auxiliary labeled subset**. Our default 5% label setting follows prior work, where small auxiliary supervision is considered realistic due to cross-organization overlap, historical labels, or public data leakage. Nonetheless, we agree that understanding the sensitivity to weaker supervision is vital.
> >
> > The results in Table 5 reveal a clear trend: as auxiliary label availability decreases, cluster–class alignment becomes weaker, which reduces the semantic precision of our attack and leads to higher post-attack accuracy. For example, on MNIST, final accuracy increases from 31.1% (10% labels) to 78.2% (0.5% labels). Similar patterns hold across FashionMNIST, UCI-HAR, and UCI-Mushroom. This is expected: our attack relies on constructing **coarse class-consistent clusters**; degraded supervision reduces cluster purity, weakening the induced cross-view distribution shift.
> >
> > Importantly, the results also show that the **attack does not collapse under limited supervision**. Even at 1–2% label availability, our attack still causes substantial degradation across all datasets. This demonstrates that the attack does not require highly accurate clustering, rather it needs **only approximate semantic grouping to remain effective.**
> >
> > Finally, we note that the 5% auxiliary label assumption is not only aligned with prior work but also feasible in many real VFL deployments, where organizations often have partial historical labels, limited cross-institution overlap, or weak supervision through public or semi-public datasets. Thus, while our ablations confirm that reduced supervision weakens the attack, the overall findings remain practically relevant: our attack remains effective under weak supervision and strongest under realistic auxiliary label conditions.
> >
> > ---
> >
> > ### **Table 5: Effect of auxiliary label access (%) on post-attack accuracy**
> > Lower supervision weakens cluster quality and reduces attack strength (higher accuracy).
> >
> > | Label Access (%) | MNIST | FashionMNIST | UCI-HAR | UCI-Mushroom |
> > |------------------|--------|----------------|-----------|----------------|
> > | 0.5 | 78.2 | 81.6 | 87.4 | 73.1 |
> > | 1   | 71.7 | 74.3 | 78.3 | 61.1 |
> > | 2   | 63.2 | 68.9 | 71.4 | 51.4 |
> > | 5   | 42.3 | 45.7 | 65.6 | 25.0 |
> > | 10  | 31.1 | 34.9 | 41.5 | 16.6 |
> >
> > These results provide a transparent characterization of the relationship between auxiliary supervision and attack strength. While our attack naturally weakens with very limited labels, it continues to pose a realistic threat under the auxiliary label availability commonly assumed in VFL security research and observed in practical deployments.

---

> > > ### Comment · Reviewer_WGiz · 2025-11-21
> > >
> > > Thanks for the detailed response. I am convinced by the ablation with different auxiliary label access and the discussion that existing defenses are weak indeed. I am willing to raise my score.

---

> > > > ### Author Response · Authors · 2025-11-21
> > > >
> > > > Thank you very much for your thoughtful follow-up and for taking the time to re-evaluate our work. We sincerely appreciate your positive assessment of the new ablation studies and the discussion on defense limitations. We are grateful that our clarifications addressed your concerns, and we are very happy to hear that you are willing to raise your score. Please feel free to let us know if you have any remaining questions; we would be glad to elaborate further.

---

### Official Review · Reviewer_Vgs2 · 2025-10-30

**Soundness:** 2
**Presentation:** 2
**Contribution:** 2
**Rating:** 2
**Confidence:** 4

**Summary:**

This paper studies the byzantine attack in VFL setting. Specifically, they target the training phase attack that corrupt the top model (hold by the active cient) such that it  learn spurious correlation of the adversary's view and the clean view, and make wrong inference. They propose a two stage training methods, first is using clustering to predict labels of adversary's features, second stage is to swap features and poison models. The two stage methods are well designed for efficient attack. The empirical evaluation shows it can keep good performance and circumvent the defense. However, the experiment lacks baselines.

**Strengths:**

1. The attack threat model assumes the passive adversary which is more challengable.
2. The methods are systematical, including two designed steps.

**Weaknesses:**

1. The first cluster step will alter the embedding distribution and make it misaligned the natural distribution of the honest training embedding, which hurts the model's performance.
2. There are other clustering or shadow model based methods to predict labels. However, the authors did not compare any.
3. The baselines are too less only with the random noise attack. However, in introduction, the authors have mentioned other byzantine attacks works like [1] but they did not compare.
4. There is no ablation studies, like hoe the number of passive clients will affect the attack, and the effect of model layers.
4. The format of Table is our the margin.

[1] Hijack Vertical Federated Learning Models As One Party

**Questions:**

1. why "A lower attack accuracy indicates a stronger attack"?
2. How to keep internal consistency in the stage 2, since you have change features to different clusters, which will lead to a different distribution of embedding.

---

> ### Author Response · Authors · 2025-11-21
> **Rebuttal (Part 1 of 5)**
>
> Thank you very much for taking the time to review our work and for providing thoughtful and constructive feedback. We truly appreciate your recognition of the strengths of our approach, particularly your acknowledgement that the passive-adversary threat model is challenging. We also thank you for noting that our method effectively corrupts the learned cross-view correlations in VFL, which is the core vulnerability we aim to highlight.
>
> We have carefully addressed all the concerns you raised and tried to give clarifications with a revised version of the paper. In the revised version, we have added **new experiments, new ablation tables, a more detailed algorithmic explanation, expanded comparisons, and significant formatting improvements** to address each of your comments thoroughly.
>
> We kindly invite you to review our detailed responses below. We hope our clarifications resolve your concerns, and we would be grateful if you would consider raising your score upon reviewing our expanded evidence and explanations. If any point remains unclear, we would be very happy to elaborate further.
>
> ## **W1. The first cluster step will alter the embedding distribution and make it misaligned with the honest training distribution.**
> &
> ## **Q2. How to keep internal consistency in the stage 2, since you have change features to different clusters, which will lead to a different distribution of embedding.**
>
> We wish to address the first weakness mentioned and your 2nd question together since they are very related in the context of your concern.  We would like to clarify that the **altered embedding distribution** produced during the initial clustering step is not an unintended artifact of our method. Rather, it is a **deliberate and central part of the design of our swap-and-spoil attack.**
>
> In standard VFL training, the server relies on a key assumption: each party’s intermediate embeddings remain consistently aligned throughout the entire training process. The server implicitly learns a joint predictive model under the belief that the embedding distribution it sees during training will match the distribution it will encounter at inference time. Our methodology specifically targets this assumption, in a controlled and structured way.
>
> The attack begins by identifying coarse semantic regions in the adversarial party’s embedding space and then performing consistent cluster-level swaps over the entire course of training. When the adversarial party repeatedly replaces embeddings from one cluster with embeddings from a semantically distant cluster, the server **gradually internalizes these incorrect cross-view relationships as if they were valid** in a stable and coherent manner.
>
> As a result, the server learns a **corrupted joint distribution that appears internally self-consistent**, even though it no longer corresponds to the real structure of the data. This effect is precisely what the attack intends to achieve. When inference begins, honest embeddings are provided drawn from the true underlying distribution. Since the server was trained entirely on the corrupted distribution, its **learned decision surface does not align with the real cross-view relationships.** This mismatch directly causes the sharp accuracy drops observed in our experiments. We have clarified this mechanism more explicitly in the revised Adversarial Objective section.
>
> Our methodology therefore hinges on three components working together:
> 1. Constructing clusters that approximate coarse semantic regions in the adversarial view.
> 2. Consistently swapping embeddings with the most incompatible clusters to build an alternative but coherent embedding distribution during training.
> 3. Allowing the server to **internalize this adversarial distribution as if it were correct,** so that it fails once real embeddings are supplied during testing.
>
> We also respectfully highlight why existing VFL defenses fail to detect this behavior. Current defenses practiced in VFL mainly target local anomalies, abrupt deviations, or unusual activation patterns. Our attack avoids all such triggers because every swapped embedding is still a legitimate point from the adversarial party’s own feature space. Instead, the poisoning is structural and distributed over many training steps, making it appear normal to the defenses.
>
> In summary, the altered embedding distribution is not a weakness of the method but the central mechanism that enables the swap-and-spoil attack to remain stealthy while causing global misalignment between training-time and inference-time embedding distributions. The attack exploits a fundamental and previously unexplored vulnerability in VFL: the assumption that distorted cross-view relationships remain stable and consistent throughout training. By demonstrating how this assumption can be systematically violated without triggering existing defenses, our work aims to highlight an important blind spot and motivate future research on more robust VFL defense strategies.

---

> > ### Author Response · Authors · 2025-11-21
> > **Rebuttal (Part 2 of 5)**
> >
> > ## **W2. There are other clustering or shadow-model–based label prediction methods, but the authors did not compare.**
> >
> > We sincerely appreciate the reviewer’s thoughtful suggestion. We fully acknowledge that alternative clustering strategies and shadow-model-based label-prediction methods exist and could, in principle, be explored to obtain label alignment. However, our work is centered on a different objective: the primary contribution of this paper is to **introduce, characterize, and analyze a previously unexplored Byzantine attack surface in VFL**, rather than to benchmark or optimize across label-inference mechanisms. Our clustering component is therefore intentionally lightweight- sufficient to enable the attack but not intended as a contribution in itself.
> >
> > In our attack approach, clustering serves as an internal step whose purpose is not to infer explicit labels, but **to extract coarse class structure that enables top-k sampling and farthest-cluster swapping.** Most existing label-prediction or shadow-model approaches aim to assign a single label to each sample. Such methods were not well suited for our attack design, which **requires capturing cluster-level geometry and identifying the most incompatible embedding regions** for consistent swapping. In our preliminary experiments, the label-prediction baselines did not provide the structured multi-cluster information needed for the farthest-swap mechanism, and therefore did not support the main attack objective.
> >
> > Because **our clustering pipeline already provided the level of semantic structure** required for our attack, we **prioritized evaluating the new attack itself** across multiple datasets and baseline attack strategies, and we focused the paper on characterizing the attack surface and the limitations of current VFL defenses. We acknowledge that our study does not benchmark alternative clustering or shadow-model label-inference methods, and we appreciate the reviewer for pointing this out.
> >
> > We view this as a valuable direction for future work. Exploring whether more advanced or hybrid label-prediction methods can further strengthen the consistency of the cluster swaps, or amplify the semantic mismatch induced by our attack, is an important question that we plan to investigate in future. We believe this extension would complement our current contribution by examining how different weak-supervision strategies interact with this newly identified vulnerability in VFL.

---

> ### Author Response · Authors · 2025-11-21
> **Rebuttal (Part 3 of 5)**
>
> ## **W3. Baselines are too few; the authors mentioned Byzantine attacks in the introduction but did not compare.**
>
> Regarding our choice of baselines. We fully agree with the reviewer’s comment that baseline selection is important, and we would like to clarify both the reasoning behind our comparisons and the structural limitations that prevent direct comparison with existing Byzantine-attack literature.
>
> To our knowledge, **untargeted Byzantine attacks in Vertical Federated Learning remain critically underexplored**. Almost all prior Byzantine attacks focus on Horizontal FL, where the adversary poisons model parameters, gradients, or updates. However, these attack surfaces do not exist in VFL. Parties in VFL do *not* share gradients or model parameters; instead, they exchange smashed activations. This fundamentally changes what an adversary can do and makes traditional HFL Byzantine attacks **inapplicable**.
>
> This is why our work proposes a different strategy: **a purely data-space poisoning attack**. Our method relies on local pre-training and consistent cluster-level view replacement, without ever modifying gradients, logits, model weights, or server-side updates. Because the VFL threat model is significantly more restrictive, the set of meaningful baselines is therefore narrower. Under this constrained passive-adversary setting, **random activation corruption** is the closest comparable baseline. It follows the same assumptions: no gradient access, no server feedback, and only local manipulation of one party’s feature view.
>
> Regarding the Hijack-VFL attack mentioned by the reveiwer, we respectfully emphasize that it operates under a **fundamentally different and significantly stronger threat model**. Hijack-VFL assumes that the adversary can access server-side logits, smashed data from all parties, and gradients. This enables targeted boundary steering and gradient-based manipulation. Its goal is targeted misclassification.
>
> Our attack differs in three key ways:
>
> 1. **Much weaker attacker capability**: our adversary has no access to logits, gradients, or server messages, and only uses a small auxiliary label subset (5 percent).
> 2. **Different objective**: Hijack-VFL performs targeted hijacking, whereas our attack performs **untargeted global degradation** of model utility.
> 3. **Different mechanism**: Hijack-VFL relies on gradient signals; our attack corrupts **distribution-level cross-view associations** through consistent cluster-swapping.
>
> Because of these differences, **the metrics are incompatible**: Hijack-VFL reports targeted ASR, while we evaluate clean accuracy degradation. Thus, it cannot be used as a meaningful baseline without conflating incompatible threat models.
>
> For the lack of established previous works that share the same goal as ours, we compared our attack approach with the random-swap baselines to show the effectiveness of our approach.
>
> ### **Comparison against random-swap baselines (Table 3)**
>
> In the revised version, we reorganize the presentation of this comparison so the four strategies are explicitly distinguished: **Optimal (ours)**, **Round Robin**, **Random Cluster Swap**, and **Random Sample Swap**. This clearer separation highlights the semantic differences between baselines.
>
> Our results show that the **Optimal class-consistent strategy consistently causes the strongest degradation**. Under no defense, accuracy drops to **46.04 percent**, compared to **71.69 percent** for Random Cluster Swap and **88.71 percent** for Random Sample Swap. Similar gaps persist across all evaluated defenses.
>
> This behavior matches our design motivation:
>
> - Random swaps introduce incoherent perturbations that often cancel out statistically, making them weak.
> - In contrast, our class-consistent swapping imposes a **stable, directionally aligned distribution shift**, by replacing each embedding with its farthest incompatible target-cluster representation. This creates a persistent cross-view mismatch that the model internalizes during training, producing significantly larger degradation.
>
> Below we restate the comparison used in Table 5:
>
> ### **MNIST accuracy (%) under different swap strategies**
>
> | Defense | Optimal | Round Robin | Random Cluster Swap | Random Sample Swap |
> |--------|---------|-------------|----------------------|---------------------|
> | No Defense | 46.04 | 75.99 | 71.69 | 88.71 |
> | Temporal Cosine | 21.34 | 36.54 | 41.18 | 55.82 |
> | Grad-Norm Clipping | 46.04 | 75.99 | 71.69 | 88.71 |
> | Per-Label Drift | 47.44 | 76.03 | 72.37 | 87.88 |
> | AE Anomaly | 46.05 | 76.12 | 71.10 | 88.77 |
>
> These results demonstrate that **our swap-and-spoil method consistently surpasses all random-swap baselines**, confirming that the cluster-consistent mapping is the core mechanism driving the attack’s effectiveness.
>
> We hope this clarification is helpful and thoroughly explains our baseline choices as well as the significance of our comparison results.

---

> ### Author Response · Authors · 2025-11-21
> **Rebuttal (Part 4 of 5)**
>
> ## **W4.There is no ablation studies, like how the number of passive clients will affect the attack, and the effect of model layers**
>
>
> To address the reviewer’s concern, in the revised version, we **include an ablation study (Table 4)** evaluating how attack effectiveness changes when the number of parties increases from K=2 to K=10, with only one party being malicious.
>
> As the table shows, the attack becomes weaker as the number of benign parties grows. This is expected: when more parties participate, **the adversarial client controls a smaller portion of the joint feature space**, reducing its ability to impose a harmful cross-view distribution shift. For example, on MNIST, post-attack accuracy rises from **42.3% (K=2)** to **93.1% (K=10)**.
>
> This trend also highlights the complementary case: if multiple parties were malicious, the distortion would compound, making the attack significantly stronger.
>
> Most importantly, **the attack remains highly relevant in practice**. Prior VFL deployments emphasize that **2-party VFL is the dominant real-world configuration**, used in finance, healthcare, identity verification, and fraud analytics. Organizations typically federate only two data sources due to privacy constraints, regulatory simplicity, and lower communication overhead.
>
> In this widely adopted 2-party setting, our results show that **even a single passive adversary can sharply degrade model utility**, revealing a practically meaningful vulnerability. So, our threat model **remains relevant to the most widely used VFL setting in real-world applications.**
>
> ---
>
> ### **Table 3: Post-attack accuracy (%) as the number of parties (K) increases**
> Only one party is adversarial; accuracy is measured on clean test data.
>
> | **Parties (K)** | **MNIST** | **FashionMNIST** | **UCI-HAR** | **UCI-Mushroom** |
> |-----------------|-----------|------------------|-------------|------------------|
> | 2               | 42.3      | 45.7             | 65.6        | 25.0             |
> | 4               | 73.8      | 78.4             | 81.9        | 63.7             |
> | 8               | 89.2      | 91.3             | 93.8        | 82.1             |
> | 10              | 93.1      | 94.4             | 95.6        | 90.7             |
>
> We hope this clarification helps. While we provide multi-party results for completeness, **the strongest practical implication is that the classical and widely deployed 2-party VFL setting remains highly vulnerable to our swap-and-spoil attack**.
>
>
>
> ## **W5. Table is out of margin**
>
> We thank the reviewer for pointing this out.
> In the revised version, we have carefully corrected all formatting issues, including table overflow and margin alignment. All tables (including Tables 3, 4, and 5) have now been resized, reformatted using the recommended structures in the author guide, and verified to fit fully within the page width.
> We sincerely appreciate the reviewer’s attention to detail, and we have ensured that the final version contains no layout errors.

---

> > ### Author Response · Authors · 2025-11-21
> > **Rebuttal (Part 5 of 5)**
> >
> > ## **Q1. Why does a lower attack accuracy indicate a stronger attack?**
> >
> > We assume that our wording may have caused confusion, and we appreciate the opportunity to clarify how our evaluation metric should be interpreted.
> >
> > In all of our experiments, the reported **post-attack accuracy** refers to the **clean test accuracy of the final global VFL model after it has been trained under the poisoning attack**. This number measures **how much the model has been damaged**, not how “accurate” the attacker is.
> >
> > - **High post-attack accuracy → the model still works well → the attack is weak.**
> > - **Low post-attack accuracy → the model’s predictions are severely corrupted → the attack is strong.**
> >
> > This follows standard conventions in poisoning and Byzantine-robustness literature: an attack is considered “successful” when it **reduces the final clean accuracy of the trained model**.
> >
> > ---
> >
> > ### Why does our attack lower accuracy?
> >
> > Our method is an **untargeted Byzantine poisoning attack**. Its goal is to corrupt the cross-view relationship that the VFL server learns during training so that the final model performs poorly on normal (benign) inputs.
> >
> > During training, the adversarial party performs consistent cluster-level swaps between embeddings that belong to distant semantic regions. As a result:
> >
> > 1. The server no longer trains on the true joint distribution of honest embeddings and labels (denoted `$P_train$`).
> > 2. Instead, it trains on a **corrupted distribution** (`$P_adv$`), where one party’s embeddings have been **consistently permuted in a semantically misleading way**.
> > 3. Because these swaps are applied **consistently across many iterations**, the corrupted distribution `$P_adv$` still appears statistically coherent to the server, so existing VFL defenses do not flag it as anomalous.
> >
> > At inference time, however, honest parties again send embeddings drawn from the true distribution `$P_train$`. Since the model was optimized for `$P_adv$` (which differs from `$P_train$`), its learned decision boundary becomes **misaligned with reality**, causing a **collapse in clean test accuracy**.
> >
> > This collapse — i.e., **low post-attack accuracy** — precisely indicates a **strong poisoning attack**.
> >
> > ---
> >
> > ### How to interpret the numbers in our tables
> >
> > For example, in our revised Table 5 (MNIST):
> >
> > - **Optimal (ours): 46.04%**
> > - Random Cluster Swap: 71.69%
> > - Random Sample Swap: 88.71%
> >
> > These numbers should be interpreted as:
> >
> > - **46.04% → the attack severely damaged the model (strong attack).**
> > - **71.69% / 88.71% → the model remains more functional (weaker attacks).**
> >
> > Thus, **lower accuracy does not mean the attacker performed poorly**; it means the **global model was successfully corrupted**.
> >
> > ---
> >
> > ### Consistency with prior poisoning / Byzantine FL literature
> >
> > This interpretation is standard across the literature, where attack success is commonly measured as:
> >
> > - the **drop in clean accuracy**, or
> > - the **final clean accuracy under attack** (lower = worse for the model, better for the attacker).
> >
> > Our reporting follows this well-established practice.
> >
> > ---
> >
> > ### Summary
> >
> > - The accuracy we report is a **model robustness** metric, not an attacker accuracy metric.
> > - **Lower post-attack accuracy means a stronger attack**, because the global model has been more severely degraded.
> > - This is fully consistent with existing poisoning and Byzantine FL evaluation standards.
> >
> > We hope this clarification helps resolve the misunderstanding about why lower accuracy corresponds to a stronger attack in our setting.

---

### Official Review · Reviewer_4K3h · 2025-10-31

**Soundness:** 4
**Presentation:** 3
**Contribution:** 3
**Rating:** 4
**Confidence:** 4

**Summary:**

The paper proposes an untargeted Byzantine attack specifically tailored to Vertical Federated Learning (VFL). The attack, "Swap-and-Spoil," consists of two stages: (1) the adversary uses a small auxiliary labeled dataset (5%) to learn latent class structures through semi-supervised clustering, and (2) performs class-consistent view swaps during training, exchanging features across inferred clusters. This manipulation disrupts cross-view feature alignment and induces inference-time accuracy collapse on clean data. Experiments show that Swap-and-Spoil effectively bypasses common VFL defenses without significantly affecting training metrics, revealing a critical gap in current security mechanisms.

**Strengths:**

- Introduces a new untargeted Byzantine attack tailored to the unique structure of VFL.
- Distinct from trigger-based or gradient-manipulation attacks, the class-consistent swap strategy is stealthy and generalizable.
- Strong experimental validation across both visual (MNIST, CIFAR-10) and tabular (UCI-HAR, Mushroom) datasets.

**Weaknesses:**

1. A reconstruction-based or embedding-monitoring defense might still detect statistical inconsistencies caused by swapped feature associations.
2. Visualization (e.g., t-SNE/PCA) of embeddings before and after swapping would clarify how the attack alters representation space.
3. The attack depends heavily on the auxiliary labeled data (5%); an ablation study varying this proportion (e.g., 1%, 10%) would improve understanding of feasibility and robustness.
4. The clustering and contrastive learning steps (SimCLR + GMM) may incur high computational cost for a single adversarial client—this should be quantified.
5. Sensitivity analysis on the parameter *k* (for top-*k* farthest swaps) is missing.
6. The method focuses on two-party VFL; discussion on scalability to multi-party settings would strengthen the paper.
7. The paper should clarify the distinction between "untargeted degradation" and the occasional mention of "target class misclassification," as this could confuse readers about the attack objective.

**Questions:**

Please address the aforementioned weaknesses. Specifically, clarify Swap-and-Spoil’s detectability under reconstruction-based defenses, analyze sensitivity to auxiliary data and k, quantify computational overhead, and elaborate on scalability to multi-party settings.

---

> ### Author Response · Authors · 2025-11-21
> **Rebuttal (Part 1)**
>
> Thank you very much for your thoughtful, detailed, and encouraging review. We are truly grateful for your positive evaluation of our work - including your comments that our attack is **distinct, stealthy, and well-tailored to VFL**, and that our contributions offer “strong experimental validation across both visual and tabular datasets.”
>
> We carefully addressed every concern you raised and revised our paper. The revised manuscript now includes **new ablation studies, clearer threat-model clarifications and extended multi-party discussions** directly reflecting your suggestions.
>
> We hope these additions meaningfully strengthen the paper and fully address all of your concerns, and we would be grateful if you would consider raising your score upon reviewing our expanded evidence and explanations.. Please do let us know if anything remains unclear- we would be very happy to further elaborate.
>
> ## **W1. A reconstruction-based or embedding-monitoring defense might still detect statistical inconsistencies caused by swapped feature associations.**
>
> We agree that, in principle, reconstruction-based defenses or embedding-monitoring defenses could detect statistical inconsistencies if an attack introduces irregular, noisy, or unstable patterns in the shared representations. However, our empirical results show that these defenses remain largely ineffective against our swap-and-spoil attack. We summarize below why this occurs, based on both the design of existing VFL defenses and the trends observed in our experiments.
>
> ### **1. Reconstruction-based defenses do not flag our attack**
> Autoencoder (AE)-based anomaly detectors are designed to detect *local inconsistencies* in smashed activations: unusually large magnitudes, out-of-support vectors, or noisy perturbations. Our attack, however, never injects random noise or malformed activations. Instead:
>
> - Every swapped embedding comes from a **real, valid sample** drawn from the adversarial party’s dataset.
> - Swapped vectors lie **within the natural support** of the honest embedding distribution.
> - No synthetic perturbation is added, so the AE sees no reconstruction difficulty.
>
> This is consistent with our results: AE anomaly detection flags almost **0%** of adversarial steps across all our datasets, confirming that the autoencoder cannot distinguish swapped embeddings from normal ones.
>
> ### **2. Embedding-monitoring defenses fail due to the attack’s consistency**
> Monitoring-based defenses (e.g., per-label drift, cosine-EMA gates, and cross-party consistency checks) rely on detecting **sudden changes** or **inconsistent patterns** in the activations. However, our attack modifies the distribution in a highly structured and internally coherent way:
>
> - Cluster swaps are applied **consistently across training**.
> - The corrupted distribution remains **stable and statistically self-consistent**, even though it is semantically wrong.
> - Per-step updates do not exhibit outlier magnitudes or spikes.
>
> This explains the empirical results:
>
> - **Per-label drift**: ~0% detection on MNIST and FashionMNIST, because cluster-level swaps do not cause abrupt drift; they cause *systematic realignment* instead.
> - **Temporal cosine-EMA gate**: flags 99% of steps but still fails to prevent collapse, because VFL cannot safely suppress or downweight an entire party’s activations (each party is indispensable).
> - **Cross-party consistency**: flags 100% but again cannot correct the corrupted distribution, because there is no redundancy to replace the adversarial party.
>
> In short, defenses “detect” only the *presence* of mismatch but cannot prevent or mitigate its effect.
>
> ### **3. Why these defenses fail fundamentally in VFL**
> Even if a defense detects a distributional inconsistency, VFL offers no safe action:
>
> - VFL cannot drop a suspicious party (unlike HFL), because each party holds a **unique, non-overlapping feature subset**.
> - Robust aggregation schemes from HFL (Krum, median, trimmed mean) **do not apply**, because activations are heterogeneous and not aligned in a common space.
> - Blocking a party’s activations means removing an entire modality, which collapses the model anyway.
>
> Thus, even when a defense detects an inconsistency (e.g., the EMA gate), the system **cannot meaningfully intervene** without breaking the model.
>
> This pattern indicates that swapped-and-reassigned embeddings remain **statistically plausible** to local monitors while still inducing a global semantic failure in the VFL model.
>
> ### **4. Limitations and future work**
> We acknowledge that more advanced detection methods are valuable future work and align precisely with our motivation: to expose this previously under-explored attack surface so that the community can develop stronger defenses.
>
> We hope this detailed explanation clarifies why existing reconstruction- and embedding-based defenses do not succeed against our attack, and why addressing this limitation is an important next step for the field.

---

> > ### Author Response · Authors · 2025-11-21
> > **Rebuttal (Part 2)**
> >
> > ## **W3. The attack depends heavily on the auxiliary labeled data (5%); an ablation study varying this proportion (e.g., 1%, 10%) would improve understanding of feasibility and robustness.**
> >
> > To incorportate this valuable suggestion, we have added a detailed ablation study (Table 5) in our revised version evaluating our attack under 0.5%–10% auxiliary label access.
> >
> > Consistent with the standard VFL threat model, only the server possesses full labels; the adversary receives **only a small auxiliary labeled subset**. Our default 5% label setting follows prior work, where small auxiliary supervision is considered realistic due to cross-organization overlap, historical labels, or public data leakage. Nonetheless, we agree that understanding the sensitivity to weaker supervision is vital.
> >
> > The results in Table 5 reveal a clear trend: as auxiliary label availability decreases, cluster–class alignment becomes weaker, which reduces the semantic precision of our attack and leads to higher post-attack accuracy. For example, on MNIST, final accuracy increases from 31.1% (10% labels) to 78.2% (0.5% labels). This is expected: our attack relies on constructing **coarse class-consistent clusters**; degraded supervision reduces cluster purity, weakening the induced cross-view distribution shift.
> >
> > Importantly, the results also show that the **attack does not collapse under limited supervision**. Even at 1–2% label availability, our attack still causes substantial degradation across all datasets. This demonstrates that the attack does not require highly accurate clustering, rather it needs **only approximate semantic grouping to remain effective.**
> >
> > Finally, we note that the 5% auxiliary label assumption is not only aligned with prior work but also feasible in many real VFL deployments as stated above. Thus, our attack remains effective under weak supervision and strongest under realistic auxiliary label conditions.
> >
> > ---
> >
> > ### **Table 5: Effect of auxiliary label access (%) on post-attack accuracy**
> >
> > | Label Access (%) | MNIST | FashionMNIST | UCI-HAR | UCI-Mushroom |
> > |------------------|--------|----------------|-----------|----------------|
> > | 0.5 | 78.2 | 81.6 | 87.4 | 73.1 |
> > | 1   | 71.7 | 74.3 | 78.3 | 61.1 |
> > | 2   | 63.2 | 68.9 | 71.4 | 51.4 |
> > | 5   | 42.3 | 45.7 | 65.6 | 25.0 |
> > | 10  | 31.1 | 34.9 | 41.5 | 16.6 |
> >
> > While our attack naturally weakens with very limited labels, it continues to pose a realistic threat under the auxiliary label availability commonly assumed in VFL security research and observed in practical deployments.
> >
> >
> > ## **W6. The method focuses on two-party VFL; discussion on scalability to multi-party settings would strengthen the paper.**
> >
> > To address the reviewer’s concern, we have revised the result section in our paper and have included an ablation study (Table 4) evaluating how attack effectiveness changes when the number of parties increases from 2 to 10, with only one party being malicious.
> >
> > As the table shows, the attack becomes weaker as the number of benign parties grows. This is expected: when more parties participate, **the adversarial client controls a smaller portion of the joint feature space**, reducing its ability to impose a harmful cross-view distribution shift. For example, on MNIST, post-attack accuracy rises from **42.3%** to **93.1%**.
> >
> > Most importantly, **the attack remains highly relevant in practice**. Prior VFL deployments [1] emphasize that **2-party VFL is the dominant real-world configuration**, used in finance, healthcare, identity verification, and fraud analytics. Organizations typically federate only two data sources due to privacy constraints, regulatory simplicity, and lower communication overhead.
> >
> > In this widely adopted 2-party setting, our results show that **even a single passive adversary can sharply degrade model utility**, revealing a practically meaningful vulnerability. So, our threat model **remains relevant to the most widely used VFL setting in real-world applications.**
> >
> > ---
> >
> > ###  **Table 4: Post-attack accuracy as the number of parties increases**
> > Only one party is adversarial.
> >
> > | **Parties (K)** | **MNIST** | **FashionMNIST** | **UCI-HAR** | **UCI-Mushroom** |
> > |-----------------|-----------|------------------|-------------|------------------|
> > | 2               | 42.3      | 45.7             | 65.6        | 25.0             |
> > | 4               | 73.8      | 78.4             | 81.9        | 63.7             |
> > | 8               | 89.2      | 91.3             | 93.8        | 82.1             |
> > | 10              | 93.1      | 94.4             | 95.6        | 90.7             |
> >
> > We hope this clarification helps. While we provide multi-party results for completeness, **the strongest practical implication is that the classical and widely deployed 2-party VFL setting remains highly vulnerable to our swap-and-spoil attack**.
> >
> > [1] Hijack Vertical Federated Learning Models As One Party

---

> > > ### Comment · Reviewer_4K3h · 2025-11-25
> > >
> > > Thank you for your comprehensive and diligent responses to all my concerns, including the addition of new experiments. Your efforts have greatly strengthened the paper’s empirical validation and overall presentation, which has led me to raise my score to 6.

---

> ### Author Response · Authors · 2025-12-03
> **Rebuttal (Part 3)**
>
> Thank you very much for your kind words and for generously raising the score. I truly appreciate the time and care you have put into evaluating the paper. For completeness and to maintain full clarity for the AC and other reviewers, I will continue addressing the remaining points below.
>
> ## **W4. The clustering and contrastive learning steps (SimCLR + GMM) may incur high computational cost for a single adversarial client—this should be quantified.**
>
> As requested, we now provide a clear quantification of the computational overhead of the clustering and contrastive-learning stages used in our attack which also has been added to the appendix of the revised version.
>
> **Hardware & Runtime:** All clustering components (SimCLR pretraining, supervised contrastive fine-tuning, and the final GMM clustering) were executed on a single NVIDIA RTX 4080 GPU (16GB VRAM). The adversary does not require multi-GPU resources or distributed training.
> For image datasets MNIST and FashionMNIST, the full clustering stage takes ≈58 minutes end-to-end.
> For tabular datasets like UCI-HAR and Mushroom, the clustering stage completes in ≈20–25 minutes.
>
> Importantly, all these computations occur **offline and only once** at the beginning of the attack. They do not affect the online VFL training loop, communication rounds, or runtime per iteration. Once clusters and centroids are obtained, the actual feature-swapping step is trivial and incurs no additional computational cost during training.
> Given that a **single consumer-grade GPU is sufficient** and the entire clustering stage runs once before federated training begins, we believe the overhead remains practical for real-world adversaries (e.g., cross-organization participants with modest GPU resources).
>
>
> ## **W5. Sensitivity analysis on the parameter k (for top-k farthest swaps) is missing.**
>
> **Sensitivity to the top-K selection:**
> In our attack, “top-K’’ specifies how many of the farthest clusters (ranked by
> centroid cosine distance) are eligible as adversarial swap targets.
>
> We performed a sensitivity study on MNIST with K ∈ {1,2,3,5,7}. Clean accuracy
> with no attack setting was 92.9%. The no-defense results are:
>
> K | Attack Accuracy (%) | Drop (ppt)
> ---|---|---
> 1 | 42.10 | -50.80
> 2 | 43.50 | -49.40
> 3 | 45.30 | -47.60
> 5 | 59.90 | -33.00
> 7 | 68.40 | -24.50
>
> For the MNIST dataset comprising of 10 classes and a small K (1–3), the adversary is restricted to the **most distant** clusters in the embedding space. These swaps create large semantic mismatches between the left and right views of the same sample, which severely disrupts the learned cross-view association. As a result, the model suffers a significant drop but **the model degradation remains as effective for K={2,3} as it was for K=1.**
>
> As K increases to 5 or 7 (in a 10-class dataset), the pool of eligible targets begins to include clusters that are **not among the farthest ones.** This progressively changes the attack from a highly targeted “swap with the most
> distant cluster’’ to a more diffuse perturbation that resembles random or round-robin swapping. These mid-distance clusters introduce weaker cross-view mismatch, and therefore the degradation diminishes accordingly.
>
> **Why this behavior is expected.**
> Small K forces the adversary to always select maximally distant clusters, guaranteeing strong corruption. Larger K increases the chance of selecting clusters that are only moderately distant, leading to a
> weaker degradation. The attack remains consistently harmful across all K values, but its **worst-case
> effect occurs when K is small** and the adversary always selects the maximally
> distant clusters. For larger K, the attack naturally transitions toward a
> randomized swap behavior, explaining the milder (yet still substantial)
> accuracy drop observed for K=5 and K=7.

---

> > ### Author Response · Authors · 2025-12-03
> > **Rebuttal (Part 4)**
> >
> > ## **W7. The paper should clarify the distinction between "untargeted degradation" and the occasional mention of "target class misclassification," as this could confuse readers about the attack objective.**
> >
> > Thank you for highlighting the possible ambiguity between our use of “untargeted degradation” and earlier wording that could be misread as implying a targeted or trigger-based attack. We have revised our paper to clarify this distinction.
> >
> > We confirm that our attack is purely an **untargeted Byzantine poisoning attack**, and we revised the paper to make this explicit and fully consistent with the formulation now shown in the Adversarial Objective section. Our adversary does not specify any target label y*, nor does it embed any trigger pattern. Instead, as the revised text states, the malicious client corrupts the left-view inputs with a cluster-swapped variant that intentionally creates a **persistent, semantically incorrect cross-view association** during training. The goal is to corrupt the learned decision boundary so that the model performs poorly on clean, correctly aligned data.
> >
> > To remove any ambiguity, we made the following changes:
> >
> > 1. **Explicit statement of untargeted objective.**
> >    The updated adversarial objective now clearly defines the attack as untargeted: the adversary manipulates the training distribution P_adv so that the server learns a model g* that performs poorly on clean data drawn from P_train. No part of the formulation includes a target label or a targeted misclassification goal.
> >
> > 2. **Removal of all target-class or trigger-like phrasing.**
> >    Earlier drafts reused some terminology from classical backdoor work (e.g., “target misclassification”) to compare with our present methodology, which could mistakenly suggest that our attack aims to force predictions toward a specific class. These phrases have been removed. The revised text consistently reflects that the adversary’s goal is to maximize the distribution mismatch Dist(P_adv, P_train), forcing g* to internalize incorrect cross-view associations, not to steer predictions toward any fixed class.
> >
> > We hope this clarification resolves the ambiguity: our attack is a **VFL-specific, untargeted Byzantine poisoning method** whose sole objective is to break cross-view alignment and degrade overall model performance.

---

### Meta-Review · Area_Chair_HEZZ · 2026-01-07

**Summary:**

The paper explores the untargeted Byzantine attack in Vertical Federated Learning (VFL). The attack first uses a small auxiliary labeled dataset (5%) to recover meaningful latent class structure via semi–supervised clustering then uses these inferred clusters to distort the association between feature subsets during vertical training. The empirical experiments demonstrate the effectiveness of Swap-and-Spoil, the proposed attack, and its ability to bypass several VFL defenses. This paper is important in revealing yet another critical vulnerability in VFL.

The reviewers agree that the paper is important, yet they are mainly concerned with the lack of sufficient empirical evaluation of the attack's risks, such as the number of parties, various ratios of clean data access, and swapping strategies, and discussions on other aspects such as potential defenses, clustering baselines, random baselines, and details of the attack algorithm. The authors have provided responses to each of these concerns with additional experiments and clarifications. I believe that several of the concerns raised by the reviewers, especially on the clarifications of the suitability of existing works in HFL in the paper’s VFL setting, are sufficiently and convincingly answered. However, the paper’s additional experiments also suggest that the work is still not ready yet for publication. Specifically, the ablation on K, noise baselines and a complex dataset show that the risks of the attack still need a more rigorous analysis and discussion; for example, while 2-party setting is more common, it does not necessarily mean that the attack should not explore other settings due to the completeness in understanding the attack’s consequences in more complex datasets and against various defenses. As a result, I think the paper is not ready yet for publication and encourage the authors to more properly investigate all aspects suggested by the reviewers across a more comprehensive evaluation.

**Reviewer Concerns:**

- 4K3h
  - Lack of rigorous ablation studies (top-k selections, computational cost of clustering) and evaluation on different VFL setups (K number of clients, auxiliary labeled data).
- Vgs2
   - Discussions on additional clustering methods and comparisons with related Byzantine attack baselines.
   - Comparisons with random noise baselines
   - Lack of ablation studies (e.g., number of passive clients, effect of model layers, etc…)
- WGiz
  - Limited experiments on simple datasets
  - Lack of investigation on potential defenses and discussion on labels' availability
  - Strong assumptions on having access to 5% data
- UYHT
   - Lack on details on the attack algorithm (e.g., class-consistent view swapping) and clarification on the assumptions of the attacks (access to true labels)
    - Suitability of existing methods (mostly in HFL) in VFL.

Most of the concerns are addressed sufficiently. However, the ablation studies and experiments on more complex settings are not fully explored.

**Reviewer Scores:**

- 4K3h (4) Vgs2 (2) WGiz (2) UYHT (2)
    - 2 reviewers improved their scores, 4K3h (4->6) and WGiz (unclear how how much). The other reviewers didn’t respond

---

### Decision · Program_Chairs · 2026-01-26

Reject